# A Long Time Span-Specific Emitter Identification Method Based on Unsupervised Domain Adaptation

Pengfei Liu [1,2,3], Lishu Guo [1,3,*], Hang Zhao [1,3], Peng Shang [1,3], Ziyue Chu [1,2,3] and Xiaochun Lu [1,2,3]

1 National Time Service Center, Chinese Academy of Sciences, Xi'an 710600, China; liupengfei@ntsc.ac.cn (P.L.); zhaohang@ntsc.ac.cn (H.Z.); shangpeng19@mails.ucas.ac.cn (P.S.); chuziyue@ntsc.ac.cn (Z.C.); luxc@ntsc.ac.cn (X.L.)
2 University of Chinese Academy of Sciences, Beijing 100049, China
3 Key Laboratory of Precise Positioning and Time Technology, Chinese Academy of Sciences, Xi'an 710600, China
* Correspondence: guolishu@ntsc.ac.cn

**Abstract:** Specific emitter identification (SEI) is a professional technology to recognize different emitters by measuring the unique features of received signals. It has been widely used in both civilian and military fields. Recently, many SEI methods based on deep learning have been proposed, most of which assume that the training set and testing set have the same data distribution. However, in reality, the testing set is generally used later than the training set and lacks labels. The long time span may change the signal transmission environment and fingerprint features. These changes result in considerable differences in data distribution between the training and testing sets, thereby affecting the recognition and prediction abilities of the model. Therefore, the existing works cannot achieve satisfactory results for a long time span SEI. To address this challenge and obtain stable fingerprints, we transform the long time span SEI problem into a domain adaptive problem and propose an unsupervised domain adaptive method called LTS-SEI. Noteworthily, LTS-SEI uses a multilayer convolutional feature extractor to learn feature knowledge and confronts a domain discriminator to generate domain-invariant shallow fingerprints. The classifier of LTS-SEI applies feature matching to source domain samples and target domain samples to achieve the domain alignment of deep fingerprints. The classifier further reduces the intraclass diversity of deep features to alleviate the misclassification problem of edge samples in the target domain. To confirm the effectiveness and reliability of LTS-SEI, we collect multiple sets of real satellite navigation signals using two antennas with 13 m- and 40 m-large apertures, respectively, and construct two available datasets. Numerous experiments demonstrate that LTS–SEI can considerably increase the recognition accuracy of the long time span SEI and is superior to the other existing methods.

**Keywords:** specific emitter identification; long time span; unsupervised domain adaptation; source domain; target domain; satellite navigation signal





## 1. Introduction

Specific emitter identification (SEI) refers to the process of identifying different emitters by analyzing the unique features of received radio signals [1]. In recent years, SEI has been widely used in both civilian and military fields. In cognitive radio systems, SEI can be used to verify the identity of primary users in a bid to prevent secondary users from occupying the licensed portion of the spectrum for a long time or to prevent disguisers from maliciously accessing the available spectrum [2]. In real combat environments, SEI can be used to identify enemies and friends, recognize interference sources, and provide battlefield situational awareness information [3]. Therefore, developing an advanced SEI technology is beneficial for us to further monitor and manage the interested targets.

For space-specific emitters, radio-frequency fingerprints (RFFs) can be extracted from their downlink signals and transmitted to the ground in a bid to achieve SEI. These RFFs

originate from the nonlinear distortion of modulated signals caused by the influence of high-power amplifiers carried by satellite transponders [4]. RFFs are unique and cannot be replicated due to the inability of the modern manufacturing processes to accurately match two power amplifiers [5]. RFFs can usually be obtained through three methods. The first method is to directly extract RFFs from the modulated signal. Refs. [6,7] used I/Q amplitude and phase imbalance as an example to study the accurate identification problem of different transmitters. Ref. [8] achieved physical layer hardware authentication by extracting I/Q constellation impairments. Ref. [9] proposed a novel method for the blind estimation of carrier frequency offset (CFO) in MPSK receivers, and CFO was used as an RFF to identify specific emitters. The second method is to obtain RFFs by performing various transformations on the modulated signal. Ref. [10] considered the SEI problem in both single-hop and relaying scenarios, as well as three RFF extraction algorithms based on the Hilbert spectrum were proposed. Ref. [11] studied an SEI method based on the variational mode decomposition and spectral features (VMD-SF). VMD decomposes the received signal simultaneously into various temporal and spectral modes. Different spectral features were extracted and achieved satisfactory results in SEI. Ref. [12] subjected the received signals to time-varying filtered empirical mode decomposition (tvf-EMD). Thereafter, the amplitude-frequency aggregation characteristics of the three-dimensional Hilbert spectrum projection and the bispectrum diagonal slice of the obtained intrinsic mode functions were used as the first and second features of SEI, respectively. Ref. [13] proposed a novel nonlinear dynamics approach based on multi-dimension approximate entropy (MApEn) for SEI. The RFFs extracted via this method could also achieve satisfactory results when using the simplest K-nearest neighbors classifier. Ref. [14] developed an SEI algorithm via joint wavelet packet analysis. The algorithm decomposes the signal via wavelet packet decomposition and extracts features of singular value center of gravity, instantaneous frequency distribution, and information demission. A support vector machine based on a voting mechanism was used to identify different emitters. However, both direct extraction and domain transformation require a careful design of RFFs and classifiers, which considerably increases the complexity of the task [15]. Additionally, the RFFs extracted using the above-mentioned two methods are susceptible to noise, which considerably fluctuates the recognition accuracy [16]. The third method is to use intelligent recognition technology to automatically extract RFFs from the received signal and complete end-to-end classification. Deep learning (DL) can compensate for the problems of the first two methods. In recent years, with the mature application of DL in fields such as computer vision and natural language processing, SEI methods based on DL have also been proposed one after another. Many research works showed that DL-based SEI performs considerably better than traditional methods [17]. Ref. [18] used the same inception-residual neural network structure for large-scale real-world ACARS and ADS-B signal data. The authors confirmed the ability of DL to address different types of radio signals. The classification accuracy on the two datasets exceeded 92% when the signal-to-noise ratio (SNR) was higher than 9 dB. Ref. [19] proposed a multisampling convolutional neural network (MSCNN) to extract RFFs from 54 ZigBee devices. MSCNN automatically uses multiple down-sampling transformations for multiscale feature extraction and classification. The classification accuracy is as high as 97% under the line-of-sight scenario with SNR = 30 dB. Ref. [20] used a convolutional neural network and compressed the bispectrum of received signals to identify specific emitters. This method can be used to extract overall feature information hidden in the original signals. Ref. [21] developed an efficient SEI method based on complex-valued neural networks (CVNNs) and network compression. CVNNs are used to enhance the recognition effect of specific emitters. Network compression ensures satisfactory recognition results while reducing the model's complexity. Ref. [22] supposed that the existing methods only consider the feature of signals or the feature after signal transformation. They ignored the temporal correlation of each feature and the relationship between the features. Therefore, a model named time-domain graph tensor attention network (TDGTAN) was proposed for SEI. This model offers higher accuracy and anti-interference performances for real-world

datasets. Table 1 discusses the mentioned literature. Additionally, some SEI methods based on DL are also used to solve more detailed practical problems such as few-shot SEI [23,24], unsupervised SEI [25,26], semi-supervised SEI [27,28], open-set SEI [29,30], and malicious attack recognition [31,32].

**Table 1.** Works Related to RFF Extraction Methods.

| RFFs Extraction Method | Work | Extracted RFFs/Models | Performance | Advantages | Disadvantages |
|---|---|---|---|---|---|
| Direct Extraction | [6] | I/Q Imbalance | SNR $\geq$ 28 dB, Pcc $\approx$ 100% | Lower sample size; no complex operations | Need to carefully design features and classifiers; require prior information on signal parameters |
| | [7] | I/Q Imbalance | Unknown SNR, Pcc $\approx$ 100% | | |
| | [8] | I/Q Constellation Impairments | SNR $\geq$ 15 dB, Pcc $\geq$ 98% | | |
| | [9] | CFO | Only RFFs extraction, no SEI | | |
| Domain Transformation | [10] | Hilbert Huang Transform | SNR $\geq$ 0 dB, Pcc $\geq$ 80% | Lower sample size; more domain transformation RFFs | Need to carefully design RFFs and classifiers; high computational complexity |
| | [11] | VMD-SF | SNR $\geq$ $-2$ dB, Pcc $\geq$ 80% | | |
| | [12] | tvf-EMD | SNR $\geq$ 16 dB, Pcc $\geq$ 85% | | |
| | [13] | MApEn | SNR = 15 dB, Pcc $\approx$ 95.65% | | |
| | [14] | Joint Wavelet Packet Analysis | SNR $\geq$ 0 dB, Pcc $\geq$ 83% | | |
| Deep Learning | [18] | Inception-Residual Neural Network | SNR $\geq$ 9 dB, Pcc $\geq$ 92% | Strong RFFs extraction ability; end-to-end recognition; enhanced recognition effect | Large number of training samples; label annotations; longer training time |
| | [19] | MSCNN | SNR $\geq$ 30 dB, Pcc $\geq$ 97% | | |
| | [20] | CNN | Achieves a Gain of about 3 dB | | |
| | [21] | CVNN + Network Compression | At High SNR, Pcc $\approx$ 100% | | |
| | [22] | TDGTAN | Unknown SNR, Pcc $\approx$ 100% | | |

Note: "Pcc" is the abbreviation for percentage correct classification.

Although SEI methods based on DL have been applied to real-world data, no effective solutions have been proposed for the long time span SEI problem. To our best knowledge, the vast majority of SEI works currently use data with the same distribution. Conversely, they all use data collected from the same batch. However, the more realistic situation is that the testing set often lags behind the training set in time, and its data distribution dynamically changes. Therefore, models trained using the training set may poorly perform on the testing set. Fine-tuning may be an effective method to solve the long time span SEI problem [33]. It involves freezing some parameters of the pretrained model and adjusting the deep layers of the network to meet the training and testing requirements of new data. However, the labels of the actual testing set are often unknown. The above-mentioned issues pose greater challenges to the practicality of DL-based SEI methods. Only limited research has been performed in the past on solving the long time span SEI problem: Ref. [34] developed an adaptive SEI system for the dynamic noise domain. The authors proposed a preprocessing algorithm called improving synchrosqueezed wavelet transforms by energy regularization, and an unsupervised neural network noise feature extracting GAN (NEGAN) (note that "GAN" means generative adversarial network). NEGAN can obtain clean RFFs from noisy signals and reduce dependence on dataset quality. However, the proposed signal preprocessing algorithm and NEGAN are relatively complex. Ref. [35] considered that the actual application environment is more complex than the ideal training environment and, therefore, developed an unsupervised domain adaptive-based modulation classification for overlapped signals. Additionally, the authors also transferred the model trained using the proposed method under an additive white Gaussian noise channel to a multipath channel. Ref. [36] proposed an SEI method based on deep adversarial

domain adaptation (DADA) to solve the problem that DL-based SEI methods are limited by the training scene and have poor generalization ability in the complex scene. DADA integrates deep neural networks into the domain adaptation problem of transfer learning. It can effectively enhance the recognition performance of the network for unlabeled samples under different conditions. Ref. [37] studied a deep multicore metric domain adaptation algorithm appropriate for underwater target recognition, which introduced the divergence deviation metric and multicore technology. This algorithm can utilize a large amount of sample data in the source domain to assist in training classifiers in the target domain. The robustness or adaptability of the above-mentioned methods was tested by manually adding noise. However, the actual environment may not match the modeled channel, particularly the space environment in which the satellite downlink signals are located. Therefore, the effectiveness of the above-mentioned methods still needs to be tested for long time span-specific emitter signals collected in real-world scenarios.

Recently, researchers have been deeply interested in the task of cross-modal object classification in the field of computer vision and have proposed various domain adaptive algorithms [33,38–45]. Domain adaptation is a subset of transfer learning. It can be used to migrate a model trained from the source domain to the target domain and enhance identification performance. Accordingly, we introduce domain adaptation to solve the long time span SEI problem. We classify the initial collected signals as source domain samples and the subsequent collected signals of the same type as target domain samples. A certain time span exists between the source domain samples and target domain samples, which may be short or long. In this study, we propose a long time span SEI method based on unsupervised domain adaptation and verify it on real datasets. Specifically, our work and contributions are as follows:

(1) The mathematical model and solution of the long time span SEI problem are presented for the first time. In this study, the problem is transformed into a domain adaptive problem. This idea is confirmed to be feasible through extensive experimental verification. To our knowledge, this is the first work to solve the long time span SEI problem.

(2) A novel long time span SEI method, called LTS-SEI, is proposed in this study. The framework of the LTS-SEI method includes four modules: data preprocessing module, feature extractor, classifier, and domain discriminator. It can learn features with domain invariance, interclass separability, and intraclass compactness. These features are confirmed to be effective in identifying different time span signals.

(3) A large number of satellite navigation signals are collected using a 13 m and a 40 m large-aperture antenna. We use these signals to construct two real datasets. Dataset A contains data on 10 navigation satellites and three data subsets. The time span between each data subset is 15 min, with a total time span of 30 min. Dataset B contains data on two navigation satellites and 14 data subsets. The time span between each data subset is in the range of 1–2 months, with a total time span of nearly 2 years.

(4) Through extensive experiments, the proposed LTS-SEI method is confirmed to satisfactorily perform for the long time span SEI problem and outperform the existing methods. To our knowledge, in addition to our previous work [29], this is also the first work to study SEI using such real, long time span signals.

## 2. Problem Formulation and Solution

This section describes the long time span SEI problem and explores ideas for a reasonable solution to the problem. Owing to several factors such as changes in space environment, inaccurate antenna pointing, and external interference, the trained model may not achieve satisfactory results in identifying new data collected at different time periods in the future. Additionally, new data often lack labels, so the model cannot adopt a supervised learning mechanism. Overall, our goal is to construct an SEI model with adaptive and predictive capabilities using limited data.

The real received space-specific emitter signal $x(t)$ can be represented as

$$x(t) = [s(t) * h(t) + n(t)] \cdot e^{-j2\pi f_m t} \tag{1}$$

where $s(t)$ represents the signal containing useful information, $h(t)$ represents the pulse response of the space channel, and $n(t)$ represents the noise and interference during transmission. Additionally, $f_m$ represents the frequency generated by the mixer to convert a radiofrequency signal into an intermediate frequency signal.

Inspired by domain adaptive learning, we consider the specific emitter signals collected at different time periods as data from different domains. We assume that the initial collected signal $\mathbf{X}_S = \{\mathbf{x}_1, \mathbf{x}_2, \mathbf{x}_3, \ldots, \mathbf{x}_N\} \in \mathcal{X}_S$ originates from source domain $S_D$ and $\mathbf{x} = \{x(t_1), x(t_2), x(t_3), \ldots, x(t_L)\}$, as shown in Figure 1. $L$ represents the length of each signal vector. $\mathbf{X}_S$ can obtain its label set $\mathbf{Y}_S = \{\mathbf{y}_1, \mathbf{y}_2, \mathbf{y}_3, \ldots, \mathbf{y}_N\} \in \mathcal{Y}_S$ through the feature extraction function $f(x; \theta)$ and Softmax classification function. $\theta$ and $N$ represent the learnable parameters of $f(x; \theta)$ and the number of categories of specific emitters, respectively. We assume that the new data $\mathbf{X}_T = \{\mathbf{x}'_1, \mathbf{x}'_2, \mathbf{x}'_3, \ldots, \mathbf{x}'_n\} \in \mathcal{X}_T$ collected after time interval $\Delta T$ come from the target domain $T_D$ and $\mathbf{x}' = \{x(t'_1), x(t'_2), x(t'_3), \ldots, x(t'_L)\}$. Additionally, the label set of $\mathbf{X}_T$ is unknown. The pretrained source domain classification model $f(\mathbf{X}_S; \theta_S)$ may not effectively perform to predict $\mathbf{Y}_T$ owing to various factors. This is because $S_D$ and $T_D$ have a relative deviation ($S_D \neq T_D$), which makes it difficult for $f(\mathbf{X}_S; \theta_S)$ to transfer the domain knowledge learned from $S_D$ to $T_D$. Therefore, when $S_D$ and $T_D$ are not aligned, the hyperplane learned by $f(\mathbf{X}_S; \theta_S)$ may not be appropriate for $T_D$, which results in confusion regarding the sample attributes.

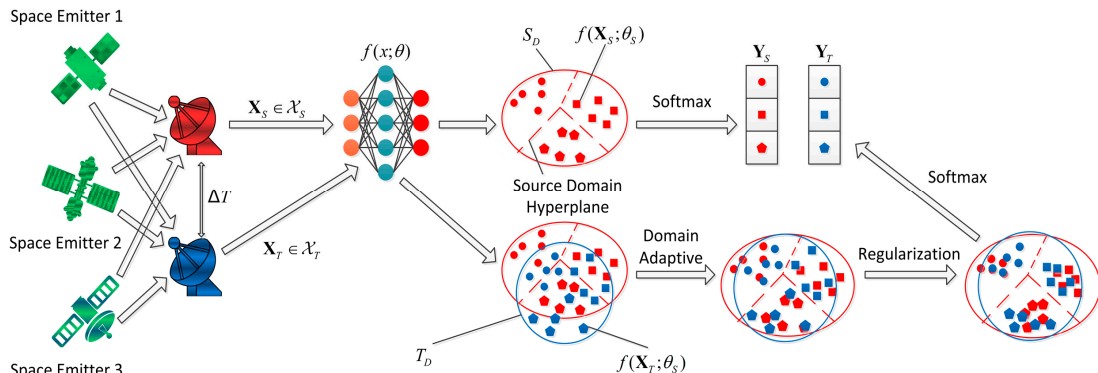

**Figure 1.** Long time span SEI problem and solution.

As previously mentioned, the labels prediction problem of specific emitter signals collected at different time periods is transformed into an unsupervised domain adaptation problem. We expect to align $S_D$ and $T_D(S_D = T_D$ or $S_D \approx T_D)$ through domain adaptation to achieve the same feature distribution for $\mathbf{X}_S$ and $\mathbf{X}_T$. If this is achieved, the source domain hyperplane can correctly predict $\mathbf{Y}_T$. Owing to the inability of domain alignment in completely eliminating the domain offset, when some samples of $T_D$ are close to the source domain classification boundary or far from the source domain feature centers, $f(\mathbf{X}_S; \theta_S)$ may experience misclassification. This problem can be alleviated using some regularization techniques to make $\mathbf{X}_S$ and $\mathbf{X}_T$ closer to their feature centers. Therefore, we expect that $f(\mathbf{X}_S; \theta_S)$ performs unsupervised learning and that the sample features of $f(\mathbf{X}_S; \theta_S)$ learning should possess the following properties: (1) Interclass separability, (2) Domain invariance, (3) Intraclass compactness.

## 3. Methodology

This section first describes the overall framework of the proposed LTS-SEI method. The various modules of the framework are then introduced in detail, including network

structure, module function, and loss. Finally, the optimization problem and optimization method of the LTS-SEI framework are presented.

### 3.1. Framework of LTS-SEI

Figure 2 shows the framework of the proposed LTS-SEI method. The framework mainly comprises four modules: (1) A data-preprocessing module to process source domain signals and target domain signals; (2) An extractor to learn the shallow RFFs of source domain samples and target domain samples; (3) A classifier to learn the deep RFFs and predict the labels of source domain samples and target domain samples; and (4) A discriminator to implement adversarial learning and a gradient reversal layer (GRL). In the LTS-SEI framework, the source domain classification model and the target domain classification model share the same feature extractor and classifier. By adding reasonable constraints, LTS-SEI tends to learn features with interclass separation, domain invariance, and intraclass compactness. Additionally, LTS-SEI adopts an unsupervised learning mechanism. The target domain samples and source domain samples are used to jointly train the entire network, and, ultimately, the classifier provides the prediction results of the target domain sample labels.

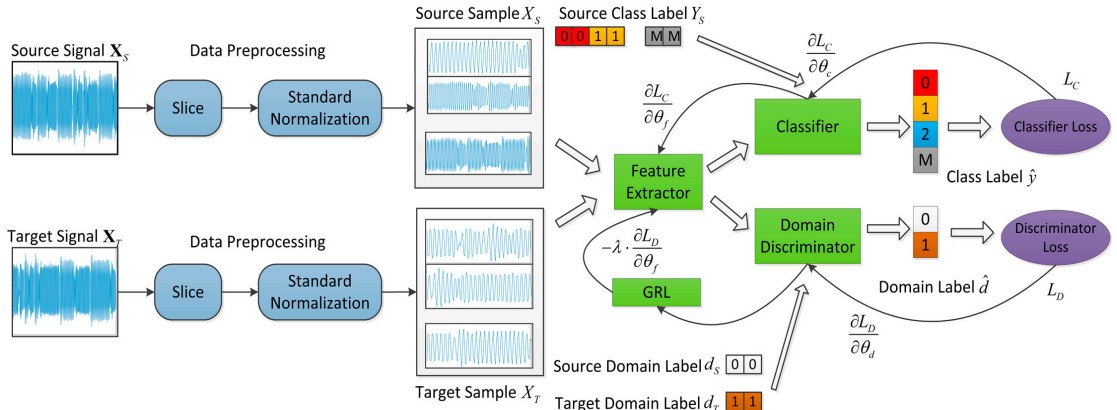

**Figure 2.** Framework of LTS-SEI.

### 3.2. Data-Preprocessing Module

The data-preprocessing module of LTS-SEI adopts the simplest processing method for the source domain signal $\mathbf{X}_S$ and target domain signal $\mathbf{X}_T$. We assume that the length of each signal vector of $\mathbf{X}_S$ and $\mathbf{X}_T$ is $L$. The source domain sample $X_S$ and target domain sample $X_T$ can be obtained through the following three steps.

(1)  We slice the source domain signal $\mathbf{X}_S$ and target domain signal $\mathbf{X}_T$ as follows:

$$X = \begin{bmatrix} x_1 & x_2 & x_3 & \cdots & x_k \\ x_{1+s} & x_{2+s} & x_{3+s} & \cdots & x_{k+s} \\ x_{1+2s} & x_{2+2s} & x_{3+2s} & \cdots & x_{k+2s} \\ \vdots & \vdots & \vdots & \vdots & \vdots \\ x_{1+\lfloor \frac{L-k}{s} \rfloor \cdot s} & x_{2+\lfloor \frac{L-k}{s} \rfloor \cdot s} & x_{3+\lfloor \frac{L-k}{s} \rfloor \cdot s} & \cdots & x_{k+\lfloor \frac{L-k}{s} \rfloor \cdot s} \end{bmatrix} \tag{2}$$

where $k$ represents length of each slice, $s$ represents slice step size, $\lfloor \cdot \rfloor$ rounding down, and $X$ represents sample matrix.

(2)  We use the following equation to standard-normalize the source domain sample $X_S$ and target domain sample $X_T$.

$$X_i = \frac{X_i - \text{mean}(X_i)}{\text{std}(X_i)}, i = 0, 1, \ldots, \left\lfloor \frac{L-k}{s} \right\rfloor \tag{3}$$

where mean($\cdot$) and std($\cdot$) are used to calculate the mean and standard deviation, respectively.

(3)  We label all source domain samples $X_S$ based on label set $\mathbf{Y}_S$.

### 3.3. Feature Extractor

Convolutional neural networks (CNNs) have been widely applied in a number of computer vision tasks, and their use has gradually extended to other fields because of their advantages of sparse connectivity and weight sharing. Many research works also confirmed that CNNs have a strong feature extraction ability in solving classic signal-processing problems such as spectrum sensing, modulation classification, and SEI. In the LTS-SEI framework, extraction of shallow features of samples also relies on convolutional modules. We designed different feature extractors for Datasets A and B, respectively, as shown in Figure 3a,b. For a 1D convolutional layer, $fi$ represents the number of filters, $ke$ represents the size of the convolutional kernel, $ac$ represents the activation function, and $st$ represents the convolutional step size. In the final layer of the feature extractor, a 1D global average pooling layer (GAP-1D) was introduced to reduce channel feature dimensions and prevent model overfitting. The following is the extraction process of the feature extractor for the source domain sample feature $F_S$, and the target domain sample feature $F_T$ can be represented as

$$f_f : X_i \in \mathbb{R}^{1\times k} \xrightarrow{\theta_f} F_i \in \mathbb{R}^{1\times fi} \tag{4}$$

where $f_f$ represents the mapping function that the feature extractor needs to learn. $\theta_f$ represents the variable parameters of $f_f$. The goal of $f_f$ is to map sample $X_i$ to feature $F_i$ that considerably impacts the recognition effect of specific emitters and is not sensitive to domain changes, as shown in Figure 2.

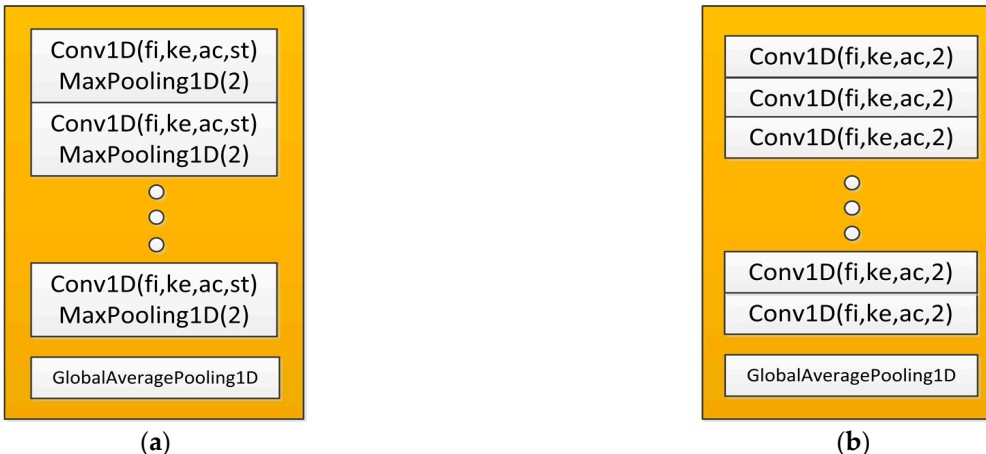

**Figure 3.** Structure of feature extractor: (**a**) Dataset A and (**b**) Dataset B.

### 3.4. Classifier

The classifier is used to integrate the shallow RFFs extracted by the feature extractor into deep features and predict labels. In the LTS-SEI framework, the classifier comprises several fully connected layers and a Softmax activation layer, as shown in Figure 4. The number of neurons of the last fully connected layer of the classifier equals the number of specific emitters. The deep feature $\mathbf{f}_i$ extracted from this layer can be represented as

$$f_c : F_i \in \mathbb{R}^{1\times fi} \xrightarrow{\theta_c} \mathbf{f}_i \in \mathbb{R}^{1\times N} \tag{5}$$

where $f_c$ represents the mapping function that the classifier needs to learn. $\theta_c$ represents the variable parameters of $f_c$. Each neuron of the last fully connected layer corresponds

to a specific emitter, and the Softmax activation layer is used to calculate the classification probability $\mathbf{p}_i$ of sample $i$ as

$$\mathbf{p}_i = \frac{\exp(\mathbf{f}_i)}{\sum_{j=1}^{N} \exp(\mathbf{f}_j)} \tag{6}$$

where $\exp(\cdot)$ is used to calculate the exponent of the feature vector. The index value that corresponds to the greatest element of $\mathbf{p}_i$ is the classifier's prediction label $\hat{y}_i$ for sample $i$. $\hat{y}_i$ can be described as

$$\hat{y}_i = \text{argmax}\{\mathbf{p}_i\} \tag{7}$$

where $\text{argmax}\{\cdot\}$ represents the index that corresponds to the maximum element value.

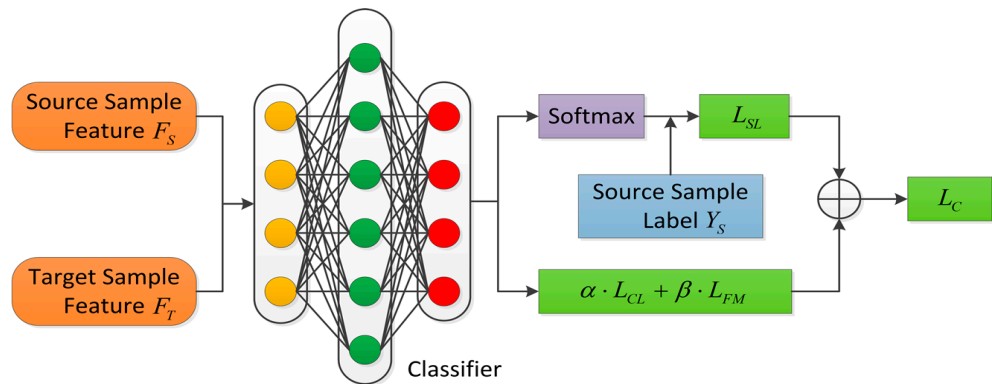

**Figure 4.** Structure of the Classifier of LTS-SEI.

The loss function should be reasonably designed at the label output end in a bid to use the chain rule and backpropagation algorithm to update the parameter $\theta_f$ of the feature extractor and the parameter $\theta_c$ of the classifier. More importantly, sample features should be interclass separated, domain invariant, and intraclass consistent. We designed an effective deep-feature learning method for the feature extractor and classifier. The method uses a hybrid metric of Softmax Loss [5,17–19], Center Loss [46], and HOMM3 Loss [40] to calculate gradient information. It can effectively reduce the distance of samples from the same specific emitters and also expand the distance of samples from different specific emitters in the feature space. Additionally, it can learn stable deep RFFs from specific emitter signals at different time periods.

(1)  Softmax Loss: Softmax Loss is often used for classification or recognition tasks in the field of signal processing. This is because its original intention is to enhance the interclass separation of features in a bid to identify different individuals. Without loss of generality, we also expect to minimize Softmax Loss to enhance the recognition performance of source domain samples. Softmax Loss can be expressed as

$$L_{SL} = -\frac{1}{M} \sum_{i=1}^{M} \sum_{j=1}^{N} y_S^{ij} \log(\mathbf{p}_S^{ij}) \tag{8}$$

where $M$ represents the number of source domain samples, and $y_S \in Y_S$ denotes the true labels of the source domain samples and is represented as a one-hot vector.

(2)  Center Loss: Center Loss was first proposed for face recognition. It can map data with intraclass diversity into feature spaces that are close to each other. Center Loss has been applied in the fields of image classification and modulation recognition to learn discriminative features. By continuously optimizing the distance between features and their clustering centers, similar samples become more compact after being mapped to the feature space. For the classifier we designed (shown in Figure 4), the source domain sample features output by the last fully connected layer are used to calculate the Center loss $L_{CL}$. Noteworthily, $L_{CL}$ comprises two parts. The first part

can reduce the distance between sample features and their class centers to enhance intraclass compactness of features. The second part will control the distance between different class centers to enhance the interclass separation of features. $L_{CL}$ can be expressed as

$$L_{CL} = \frac{1}{2}(\sum_{i=1}^{K}\left\|\mathbf{f}_S^i - \mathbf{c}_{y_i}\right\|_2^2 + \sum_{i,j=1,i\neq j}^{N} \max(0, \sigma - \left\|\mathbf{c}_{y_i} - \mathbf{c}_{y_j}\right\|_2^2)) \tag{9}$$

where $K$ represents the number of small batch samples, $\mathbf{f}_S^i$ represents the feature vector extracted from the source domain sample $X_i$ by the last fully connected layer, and $\mathbf{c}_{y_i}$ represents the class center corresponding to the real label of $X_i$. $\|\cdot\|_2^2$ is used to calculate the square Euclidean distance. $\sigma$ represents a variable parameter that controls the distance between different class centers. After each iteration, the class centers of the entire training set need to be updated because all training sample features need to be recalculated. However, this is cumbersome and impractical because the model uses only a small batch of samples to participate in training each time. Therefore, we use the mean of small batch sample features to approximate the global class centers. Batch class center $\mathbf{c}_{y_i}$ can be described as

$$\mathbf{c}_{y_i} = \frac{1}{b}\sum_{i=1}^{b} \mathbf{f}_{y_i} \tag{10}$$

where $\mathbf{f}_{y_i}$ and $b$ represent the features and their numbers of the same category as $X_i$, respectively. Since $\mathbf{c}_{y_i}$ does not train with the network parameters, we introduce $\gamma \in [0,1]$ to control the learning rate of $\mathbf{c}_{y_i}$ to prevent singular samples in the training set from causing considerable fluctuation for class centers in the feature space. The update method of $\mathbf{c}_{y_i}$ is designed as follows:

$$\mathbf{c}_{y_i}^{t+1} = \mathbf{c}_{y_i}^t - \gamma \cdot \Delta\mathbf{c}_{y_i} \tag{11}$$

$$\Delta\mathbf{c}_{yi} = \frac{\sum\limits_{j=1}^{K} \delta(y_j = y_i)(\mathbf{c}_{yi}^t - \mathbf{f}_j)}{1 + \sum\limits_{k=1}^{K} \delta(y_k = y_i)} \tag{12}$$

where $\mathbf{c}_{y_i}^{t+1}$ represents the class center updated by class $y_i$ at time $t+1$, $\mathbf{c}_{y_i}^t$ represents the class center updated by class $y_i$ at time $t$, and $\delta(\cdot)$ represents the impulse function. When the condition in the parentheses holds, $\delta(\cdot) = 1$. Otherwise, $\delta(\cdot) = 0$.

(3)  HOMM3 Loss: We added a new constraint to the deep-features output by the classifier to enhance the domain adaptation ability of the model. By optimizing the feature-matching loss, the source domain and target domain are forced to align in the deep-feature space. High-order statistics (such as high-order moments and high-order cumulants) are typically used to describe the intrinsic distribution of signals and are also commonly used as recognition features of different signals. When a random signal follows a Gaussian distribution, its statistical characteristics can be understood through a mathematical expectation (first-order statistic) and an autocorrelation function (second-order statistic). However, the expression of statistical characteristics of non-Gaussian distribution signals by low-order statistics is limited, which may impact the effectiveness of domain matching. We propose to reduce the distance between the high-order moments of the source domain sample features and target domain sample features by optimizing the high-order moment matching loss. However, the calculation of higher-order tensors introduces higher temporal and spatial complexities. When the number of neurons in the bottleneck layer is $m$, the dimension of the $p$ order statistics reaches $O(m^p)$. In a bid to reduce complexity and achieve

fine-grained domain alignment of features, we selected the last fully connected layer of the classifier as the bottleneck layer and used this layer to calculate third-order statistics of the features. At this time, the spatial complexity of the feature-matching loss in the bottleneck layer is $O(N^3)$. This order of magnitude is acceptable for the model to learn domain-invariant features. The third-order moment matching loss $L_{FM}$ can be described as

$$L_{FM} = \frac{1}{N^3} \left\| \frac{1}{K_S} \sum_{i=1}^{K_S} \phi_{\theta_S}(X_S^i)^{\otimes 3} - \frac{1}{K_T} \sum_{i=1}^{K_T} \phi_{\theta_T}(X_T^i)^{\otimes 3} \right\|_2^2 \tag{13}$$

where $N$ represents the number of categories of specific emitters, and $K_S$ and $K_T$ represent the number of small batch samples in the source domain and target domain, respectively. $\phi_{\theta_S}(\cdot)$ and $\phi_{\theta_T}(\cdot)$ represent the bottleneck layers of the source classifier and the target classifier, respectively. They all output deep features with $1 \times N$ dimensions. $\theta_S$ and $\theta_T$ represent the parameters of the bottleneck layers of the source classifier and target classifier, respectively. $X_S$ and $X_T$ represent the source domain samples and target domain samples, respectively. $\otimes$ represents a vector product operator. $\mathbf{A}^{\otimes 3}$ represents the third-order tensor power of vector $\mathbf{A}$ and can be expressed as

$$\mathbf{A}^{\otimes 3} = \mathbf{A} \otimes \mathbf{A} \otimes \mathbf{A} \tag{14}$$

Since the source domain classifier and target domain classifier share network parameters, $L_{FM}$ can be simplified as

$$L_{FM} = \frac{1}{N^3} \left\| \frac{1}{K} \sum_{i=1}^{K} \phi_{\theta_S}(X_S^i)^{\otimes 3} - \frac{1}{K} \sum_{i=1}^{K} \phi_{\theta_S}(X_T^i)^{\otimes 3} \right\|_2^2 \tag{15}$$

Additionally, because the bottleneck layer is the last fully connected layer of the classifier, $L_{FM}$ can be further denoted as

$$L_{FM} = \frac{1}{N^3} \left\| \frac{1}{K} \sum_{i=1}^{K} (\mathbf{f}_S^i)^{\otimes 3} - \frac{1}{K} \sum_{i=1}^{K} (\mathbf{f}_T^i)^{\otimes 3} \right\|_2^2 \tag{16}$$

Finally, the hybrid loss $L_C$ of the classifier can be represented as

$$
\begin{aligned}
L_C &= L_{SL} + \alpha \cdot L_{CL} + \beta \cdot L_{FM} \\
&= -\frac{1}{M} \sum_{i=1}^{M} \sum_{j=1}^{N} y_S^{ij} \log(\mathbf{p}_S^{ij}) + \frac{\alpha}{2} \left( \sum_{i=1}^{K} \left\| \mathbf{f}_S^i - \mathbf{c}_{y_i} \right\|_2^2 + \sum_{i,j=1, i \neq j}^{N} \max(0, \sigma - \left\| \mathbf{c}_{y_i} - \mathbf{c}_{y_j} \right\|_2^2) \right) \\
&\quad + \frac{\beta}{N^3} \left\| \frac{1}{K} \sum_{i=1}^{K} (\mathbf{f}_S^i)^{\otimes 3} - \frac{1}{K} \sum_{i=1}^{K} (\mathbf{f}_T^i)^{\otimes 3} \right\|_2^2
\end{aligned} \tag{17}
$$

where $\alpha$ and $\beta$ denote parameters for the weight of control Center Loss and HOMM3 Loss, respectively.

### 3.5. Domain Discriminator and Gradient Reversal Layer

In addition to using HOMM3 Loss to promote deep-feature alignment between the source domain and target domain, the LTS-SEI method also introduces adversarial learning to achieve shallow feature matching between the source domain and target domain. A domain discriminator is designed to compete with the feature extractor, as shown in Figure 2. The domain discriminator of the LTS-SEI comprises several fully connected layers and a sigmoid activation layer, as shown in Figure 5. Noteworthily, the last fully connected layer of the domain discriminator only has a neuron. The feature extractor and domain discriminator of LTS-SEI are similar to the generator and discriminator of a GAN. Unlike the generator, the goal of the feature extractor is not to synthesize samples but to generate domain-invariant features. We assume that the domain labels of the source

domain samples are 0 (true) and those of the target domain samples are 1 (false). When the domain discriminator is unable to distinguish whether the output of the feature extractor originates from the source domain or target domain, it is indicated that the feature extractor and domain discriminator have reached the Nash equilibrium in the process of mutual game. We assume that the alignment between the source domain and target domain has been achieved, and the probability of the domain discriminator output being 0 or 1 should be 0.5 each. Therefore, the purpose of the domain discriminator is to minimize the domain discrimination loss and correctly identify the domain labels of shallow features. The purpose of the feature extractor is to maximize the domain discrimination loss and confuse the judgment of the domain discriminator. Here, the domain discrimination loss $L_D$ is defined as

$$L_D = -\frac{1}{K}\sum_{i=1}^{K}[d_i \cdot \log(p_{d_i}) + (1-d_i) \cdot log(1-p_{d_i})] \tag{18}$$

where $K$ represents the number of small batch samples, $d_i$ denotes the real domain label of sample $X_i$ and is represented as a one-hot vector, and $p_{d_i}$ represents the recognition probability of sample $X_i$ by the domain discriminator.

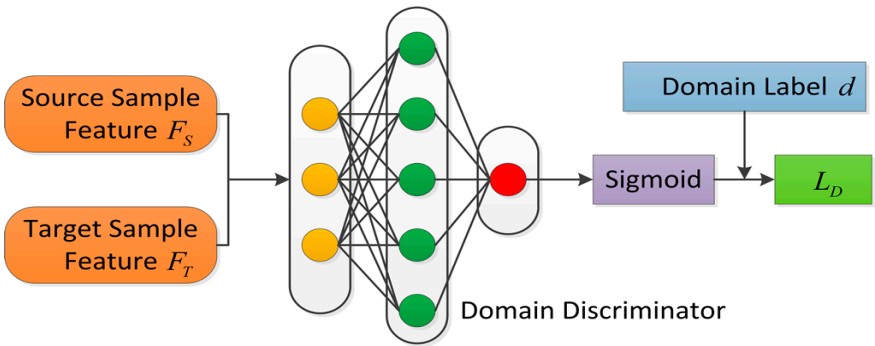

**Figure 5.** Structure of the Domain Discriminator of LTS-SEI.

The traditional neural network optimization method requires to minimize loss when updating parameters. We introduce a GRL between the feature extractor and domain discriminator because the purpose of the feature extractor is to maximize $L_D$. GRL maintains the input unchanged during forward propagation and reverses the gradient by multiplying it by a negative scalar $\lambda$ during reverse propagation. Based on this idea, the feature extractor and domain discriminator implement adversarial learning and ensure that the shallow features output by the feature extractor are domain invariant.

*3.6. Optimization Problem of LTS-SEI*

Based on the gradient propagation process of the feature extractor, classifier, and domain discriminator shown in Figure 2, we can use the network parameter $(\theta_e, \theta_c, \theta_d)$ to represent the overall classification loss $L_A$ of the LTS-SEI framework. $L_A$ is recorded as

$$\begin{aligned} L_A(\theta_f, \theta_c, \theta_d) &= L_C(\theta_f, \theta_c) - \lambda \cdot L_D(\theta_f, \theta_d) \\ &= L_{SL}(\theta_f, \theta_c) + \alpha \cdot L_{CL}(\theta_f, \theta_c) + \beta \cdot L_{FM}(\theta_f, \theta_c) - \lambda \cdot L_D(\theta_f, \theta_d) \end{aligned} \tag{19}$$

where $\theta_f$, $\theta_c$, and $\theta_d$ represent network parameters for the feature extractor, classifier, and domain discriminator, respectively.

Therefore, the LTS-SEI method seeking the optimal network parameters $(\hat{\theta}_e, \hat{\theta}_c, \hat{\theta}_d)$ is equivalent to solving the optimization problem of minimum/maximum of the overall classification loss $L_A$ as follows

$$(\hat{\theta}_f, \hat{\theta}_c) = \arg \min_{\theta_f, \theta_c} L_A(\theta_f, \theta_c, \hat{\theta}_d) \tag{20}$$

$$\hat{\theta}_d = \underset{\theta_d}{\operatorname{argmax}} L_A(\hat{\theta}_f, \hat{\theta}_c, \theta_d) \tag{21}$$

### 3.7. Optimization Method of LTS-SEI

The parameters of the feature extractor, classifier, and domain discriminator of LTS-SEI can be updated through (22), (23), and (24), respectively:

$$\theta_f^{t+1} = \theta_f^t - \mu\left(\frac{\partial L_C^t}{\partial \theta_f^t} - \lambda\frac{\partial L_D^t}{\partial \theta_f^t}\right) \tag{22}$$

$$\theta_c^{t+1} = \theta_c^t - \mu\frac{\partial L_C^t}{\partial \theta_c^t} \tag{23}$$

$$\theta_d^{t+1} = \theta_d^t - \mu\frac{\partial L_D^t}{\partial \theta_d^t} \tag{24}$$

Algorithm 1 provides the optimization and testing process of the LTS-SEI framework. When obtaining the optimal network parameter $(\hat{\theta}_e, \hat{\theta}_c, \hat{\theta}_d)$, LTS-SEI can learn features with interclass separability, domain invariance, and intraclass compactness. The multiple experiments in Section 5 indicate that these features are beneficial to solving the long time span SEI problem.

---

**Algorithm 1** The optimization and testing process of the LTS-SEI framework

---

**Input:** Source domain signal $\mathbf{X}_S$, Source domain signal label $\mathbf{Y}_S$, Target domain signal $\mathbf{X}_T$, Source domain label $d_S$, Target domain label $d_T$.
**Output:** Optimal network parameter $(\hat{\theta}_e, \hat{\theta}_c, \hat{\theta}_d)$, Target domain sample label $Y_T$.
1. Obtain source domain sample $X_S$, target domain sample, $X_T$ and source domain sample label $Y_S$. (See data-preprocessing module.)
2. Forward propagation:
(1) From feature extractor to classifier:
Input $X_S$ and $X_T$ into the feature extractor to obtain shallow features $F_S$ and $F_T$. (See Equation (4))
Input $F_S$ and $F_T$ into the classifier to obtain deep features $\mathbf{f}_S$ and $\mathbf{f}_T$, prediction
    label $\hat{y}_S$, and prediction probability $\mathbf{p}_S$ of $X_S$. (See Equations (5)–(7))
Compute Softmax Loss $L_{SL}$ using $\hat{y}_S$ and $Y_S$. (See Equation (8))
Compute Center Loss $L_{CL}$ using $\mathbf{f}_S$ and $Y_S$. (See Equation (9))
Compute HOMM3 Loss $L_{FM}$ using $\mathbf{f}_S$ and $\mathbf{f}_T$. (See Equation (16))
Compute the hybrid loss $L_C$ of the classifier. (See Equation (17))
(2) From feature extractor to domain discriminator:
Input $X_S$ and $X_T$ into the feature extractor to obtain shallow features $F_S$ and $F_T$. (See Equation (4))
Compute domain discrimination loss using $F_S$, $F_T$, $d_S$ and $d_T$. (See Equation (18))
3. Back propagation:
(1) Calculate gradient information $\frac{\partial L_C}{\partial \theta_c}$, $\frac{\partial L_C}{\partial \theta_f}$, $\frac{\partial L_D}{\partial \theta_d}$ and $\frac{\partial L_D}{\partial \theta_f}$.
(2) Update $\theta_c$, $\theta_f$, and $\theta_d$ through random gradient descent. (See Equations (22)–(24))
4. Repeat Step 2 and 3 until the maximum number of iterations is met.
5. Save the optimal parameter $(\hat{\theta}_f, \hat{\theta}_c, \hat{\theta}_d)$ after completing the LTS-SEI training.
6. Input $X_T$ into the feature extractor $(\hat{\theta}_f)$ and classifier $(\hat{\theta}_c)$.
7. Output target domain sample label $Y_T$.

---

## 4. Dataset

This section details real-world data used to validate the effectiveness and reliability of the LTS-SEI method. We collect many long time span satellite navigation signals using a 13-m and a 40-m large-aperture antenna, respectively. All signals are first processed by the data-preprocessing module of the LTS-SEI framework. We then construct two available datasets and name them Datasets A and B. The details of the two datasets are discussed in the following.

### 4.1. Dataset A

Dataset A comprises data on signal samples from 10 navigation satellites. These signals are observed using an Agilent spectrum analyzer, received using a 13-m antenna and collected using a signal-acquisition device with a 250 MHz sampling rate at the Xi'an Aerospace Base Park of National Time Service Center, Chinese Academy of Sciences. Among the 10 navigation satellites, five are from Japan's Quasi-Zenith Satellite System (QZSS) and the other five are from the Indian Regional Navigation Satellite System (NAVIC). We collected three time navigation signals with a frequency of 1176.45 MHz broadcasted by 10 navigation satellites. The collection interval is 15 min. Figure 6a,b show the spectrums of navigation signals of a QZSS satellite and a NAVIC satellite after denoising and smoothing filtering, respectively. Both spectrums were calculated and plotted using $10^6$ data points. The navigation signals broadcasted by five QZSS satellites at a 1227.60 MHz frequency were also collected because we use a broadband signal-acquisition system. However, this did not affect our SEI work, as both signals indicate the same satellite. We constructed three subsets of data using navigation signals collected three times and labeled them as A1, A2, and A3.

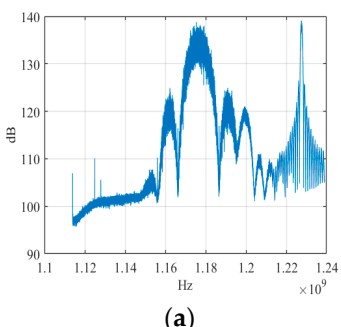
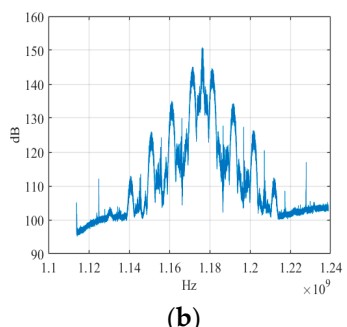
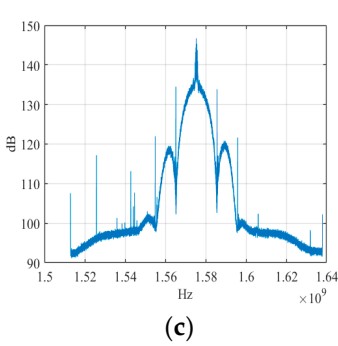

(**a**)         (**b**)         (**c**)

**Figure 6.** Spectrum of navigation signals from various satellite systems: (**a**) QZSS, (**b**) NAVIC, and (**c**) GPS.

### 4.2. Dataset B

Dataset B comprises data on signal samples from two navigation satellites of the Global Positioning System (GPS). These signals were observed using an Agilent spectrum analyzer, received using a 40 m antenna, and collected using a signal-acquisition device with a 250 MHz sampling rate at the Luonan Haoping Station of National Time Service Center, Chinese Academy of Sciences. Similar to the signal-processing method in Dataset A, Figure 6c shows the spectrum of a navigation signal broadcasted by a GPS satellite at 1575.42 MHz frequency. The spectrum was also calculated and plotted using 10 M data points. The quality of GPS satellite navigation signals collected using a 40 m large-aperture antenna was higher. We conducted 2 years of observation on two GPS satellites and collected multiple long time span signals. Specifically, we constructed 14 data subsets using these navigation signals and labeled them as B1, B2, . . . . . . , and B14, respectively. The time span between these data subsets is approximately in the range of 1–2 months, as shown in Table 2.

**Table 2.** GPS navigation signal acquisition time.

| Data Subset | Specific Emitter 1 | Specific Emitter 2 |
| :---: | :---: | :---: |
| B1 | 8 May 2021 | 8 May 2021 |
| B2 | 24 May 2021 | 22 May 2021 |
| B3 | 9 June 2021 | 24 June 2021 |
| B4 | 10 July 2021 | 6 July 2021 |
| B5 | 19 August 2021 | 16 August 2021 |
| B6 | 8 September 2021 | 5 September 2021 |

**Table 2.** *Cont.*

| Data Subset | Specific Emitter 1 | Specific Emitter 2 |
|:---:|:---:|:---:|
| B7 | 14 October 2021 | 23 October 2021 |
| B8 | 6 December 2021 | 3 December 2021 |
| B9 | 11 February 2022 | 13 February 2022 |
| B10 | 8 March 2022 | 11 March 2022 |
| B11 | 10 May 2022 | 14 May 2022 |
| B12 | 15 June 2022 | 16 June 2022 |
| B13 | 7 August 2022 | 3 August 2022 |
| B14 | 21 September 2022 | 28 September 2022 |

## 5. Experimental Results and Discussion

This section confirms the recognition performance of the proposed LTS-SEI method on two real specific emitter datasets. The experimental data, identification model, parameter settings, and evaluation criteria are first detailed for a clearer understanding of our experimental process. Subsequently, we comprehensively analyze the performance of the LTS-SEI method from different perspectives. Finally, we extend the LTS-SEI method to small samples and evaluate its effectiveness.

### 5.1. Experimental Data

The collection time of each specific emitter signal file was 2 s, including 500 M data points. In a bid to reduce memory and time consumption, we used only a small amount of data from each signal file to construct a data subset. In Dataset A, each data subset contains 12,000 samples, which means the number of samples of each specific emitter is 1200. For Dataset B, each data subset also contains 12,000 samples, which means the number of samples of each specific emitter is 6000. All signal samples have a length of 4000. During the experiment, each data subset was divided into a training set, a validation set, and a testing set in a ratio of 0.8:0.1:0.1. Based on the signal-acquisition time, A1 and B1 were used as source domain samples for Datasets A and B, respectively. Other data subsets were used as target domain samples.

### 5.2. Identification Model

The identification models were constructed using TensorFlow. They were all trained and tested on a workstation equipped with an Intel (R) Core (TM) i9-10900K CPU and NVIDIA GeForce RTX3090 GPU. Figure 7a,b show the recognition models used for Datasets A and B, respectively. The feature extractors, classifiers, and domain discriminators of the two models follow the network structure described in Section 3. The feature extractor of Model A comprises four 1D convolutional modules. Every 1D convolutional module includes a 1D convolutional layer and a maxpooling layer. The feature extractor of Model B comprises 10 1D convolutional layers. They were designed as lightweight as possible to reduce training parameters and time.

### 5.3. Parameters Setting

An appropriate parameters setting is crucial for training a superior model. Each module of the LTS-SEI framework involves multiple hyperparameters, and we must explain these parameters in detail. Table 3 presents the values of nine hyperparameters that need to be set during the training process. Additionally, we need to explain the following four points about the parameters setting:

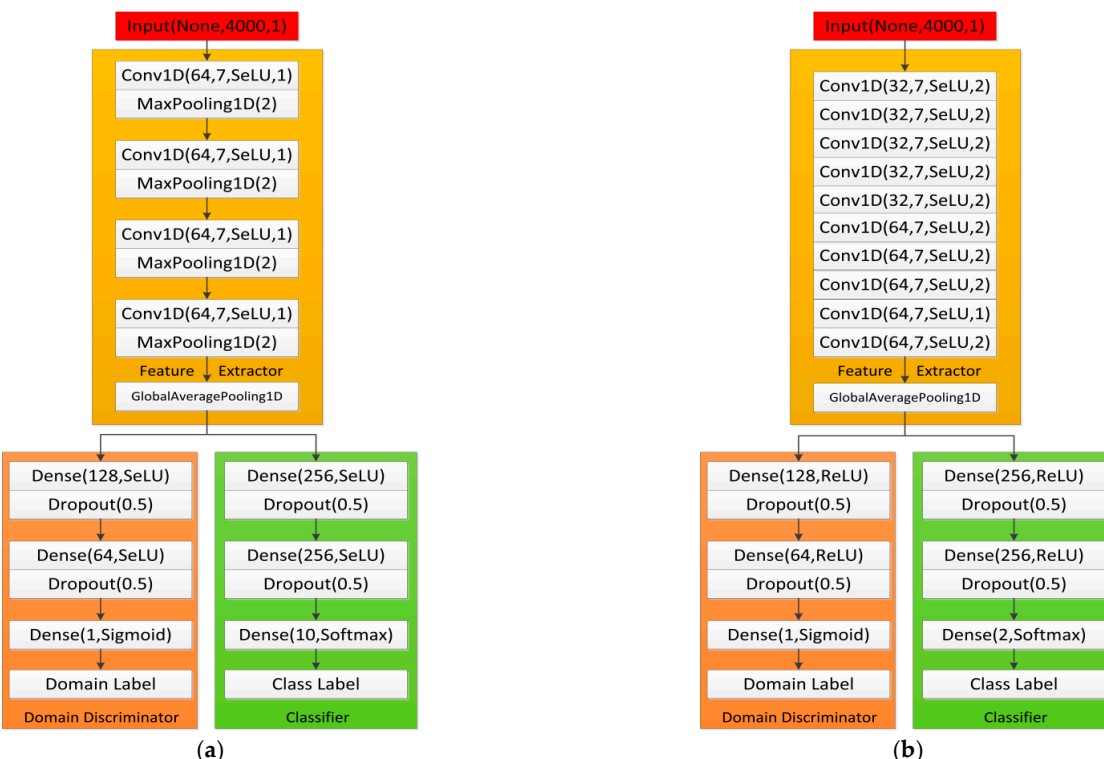

**Figure 7.** Identification models of two datasets: (**a**) Dataset A and (**b**) Dataset B.

(1)  Learning rate $\mu$: Consistent with traditional models, $\mu$ is used to control the learning rate of the feature extractor, classifier, and domain discriminator. For Dataset A, which has a short time span, Model A is easy to train. The learning rate of Model A is set to the commonly used value of 0.01. For Dataset B, which has a long time span, the learning rate of Model B gradually decreases with the increase in iterations to achieve fast convergence. $\mu$ is defined as

$$\mu = \frac{1}{\left(1 + \frac{epoch}{E}\right)} \tag{25}$$

where *epoch* represents the current number of iterations of the model and *E* the total number of iterations of the model.

(2)  Reversal Scalar $\lambda$: $\lambda$ is a weight scalar used to control negative gradient. Similar to (1), we set the reversal scalar of Model A to a constant. For Model B, as $\mu$ gradually decreases, $\lambda$ is set as an increasing function with respect to number of iterations in a bid to encourage the feature extractor to continuously learn domain-invariant features. $\lambda$ is represented as

$$\lambda = \frac{2}{1 + e^{-10 \cdot \frac{epoch}{E}}} - 1 \tag{26}$$

(3)  Epoch *E*: After multiple experimental verifications, we determined that for Data Subsets B3, B6, and B12, Model B can achieve nearly 100% accuracy through only 100 iterations. Therefore, for these three data subsets, the number of iterations of the model are set to 100. The number of iterations for other data subsets are 300.

(4)  Center Loss weight $\alpha$ and HOMM3 Loss weight $\beta$: We set the weight factors of the two loss functions to the same value to quickly determine suitable hyperparameters $\alpha$ and $\beta$. The weight range is [0, 1] and the step size is 0.001. We experimentally determined that the domain adaptation ability of the model decreases when the two weight factors are small. The model may overfitting when the two weight factors are large. Model A can maintain a high recognition accuracy on Dataset A when both

weight factors are 0.1. Model B can maintain a high recognition accuracy on Dataset B when both weight factors are 0.01.

**Table 3.** Parameter setting of LTS-SEI.

| Symbol | Meaning | Model A | Model B |
|:---:|:---:|:---:|:---:|
| $\alpha$ | Center Loss weight | 0.1 | 0.01 |
| $\beta$ | HOMM3 Loss weight | 0.1 | 0.01 |
| $l$ | Initial learning rate | 0.0007 | 0.001 |
| $\mu$ | Learning rate | 0.01 | Equation (25) |
| $\lambda$ | Reversal Scalar | 0.01 | Equation (26) |
| $E$ | Epoch | 100 | 100/300 |
| $B$ | Batch Size | 64 | 64 |
| $\gamma$ | Center learning rate | 0.5 | 0.5 |
| $\sigma$ | Marginal factor | 100 | 100 |

*5.4. Evaluation Criteria*

We adopted four evaluation criteria, namely accuracy, precision, recall, and F1-score in a bid to accurately analyze and fairly compare the effectiveness of the LTS-SEI method in addressing the long time span SEI problem. For each individual category, these criteria can be separately calculated as follows:

$$accuracy = \frac{TP + TN}{TP + FP + TN + FN} \tag{27}$$

$$precision = \frac{TP}{TP + FP} \tag{28}$$

$$recall = \frac{TP}{TP + FN} \tag{29}$$

$$F1_{score} = \frac{2 \times precision \times recall}{precision + recall} \tag{30}$$

where $TP$ represents the number of samples with both real and predicted categories being positive, $TN$ the number of samples with both real and predicted categories being negative, $FP$ the number of samples with the real category being negative and the predicted category being positive, and $FN$ the number of samples with the real category being positive and the predicted category being negative. For the multiclassification task in this study, we can obtain multiclassification evaluation criteria using the averages of accuracy, precision, recall, and F1-score:

$$acc = \frac{1}{N} \sum_{i=1}^{N} accuracy_i \tag{31}$$

$$pre = \frac{1}{N} \sum_{i=1}^{N} precision_i \tag{32}$$

$$rec = \frac{1}{N} \sum_{i=1}^{N} recall_i \tag{33}$$

$$f1 = \frac{1}{N} \sum_{i=1}^{N} F1_{score_i} \tag{34}$$

*5.5. Performance Comparison with Domain Adaptation Methods*

In this study, the long time span SEI problem is considered a domain adaptation problem. Therefore, LTS-SEI is compared with the existing domain adaptation meth-

ods, including maximum mean discrepancy (MMD) [33], central moment discrepancy (CMD) [39], higher-order moment matching (HOMM) [40], unsupervised domain adaptation by backpropagation (UDAB) [41], deep adaptation networks (DANs) [43], and joint domain alignment (JDA) [44]. The performance upper limit of all methods is the effect when the distributions of the source domain data and target domain data are the same. Additionally, shallow RFFs and deep RFFs are used to confirm the domain alignment effectiveness of these methods.

*Shallow layer*: Shallow RFFs refer to features output by the GAP-1D of the feature extractor. Table 4 presents the average recognition accuracy of the original I/Q signal (no domain adaptation), other domain adaptive methods, the proposed LTS-SEI, and the performance upper limit. Other domain adaptive methods are used for a shallow feature alignment. Evidently, all domain adaptive methods have improvements for A1->A2 and A1->A3. We suppose that although there is a 15-min time span between A1, A2, and A3, they are all signals collected from the same batch, which results in considerable similarity of data distribution. Model A still has a satisfactory predictive ability for data subsets with shorter time spans. Even without using domain adaptive methods, the original I/Q signal can achieve 85% average accuracy. Both UDAB and the proposed LTS-SEI method can reach the upper limit of prediction accuracy. Except for B3, the original I/Q signal and existing domain adaptive methods suffer from poor predictions for B2, B4, and B5. This may be due to changes in space environment or fingerprint features caused by a long time span, resulting in considerable differences in data distribution between B1 and other data subsets. Therefore, Model B loses its predictive and recognition capabilities for Data Subsets B2, B4, and B5. Despite a one-month time span between B1 and B3, LTS-SEI achieves the average accuracy similar to the performance upper limit. For B2, B4, and B5, the advantage of LTS-SEI is more obvious, with an average accuracy of exceeding 96%. Therefore, the existing methods that only align shallow features in the feature space cannot achieve satisfactory recognition results.

**Table 4.** Performance comparison between various methods (shallow layer).

| Method | Average Recognition Accuracy | | | | | |
|---|---|---|---|---|---|---|
| | **A1->A2** | **A1->A3** | **B1->B2** | **B1->B3** | **B1->B4** | **B1->B5** |
| I/Q | 0.8958 | 0.8592 | 0.5200 | 0.8650 | 0.4792 | 0.4692 |
| MMD | 0.9108 | 0.8933 | 0.5400 | 0.8458 | 0.4692 | 0.5008 |
| DAN | 0.9367 | 0.8883 | 0.5456 | 0.8708 | 0.4667 | 0.4883 |
| JDA | 0.9425 | 0.8867 | 0.5233 | 0.8542 | 0.4975 | 0.4933 |
| CMD | 0.9833 | 0.9417 | 0.5308 | 0.8467 | 0.4625 | 0.4700 |
| HOMM | 0.9842 | 0.9492 | 0.5367 | 0.8433 | 0.4842 | 0.4967 |
| UDAB | 0.9975 | 0.9925 | 0.5550 | 0.9425 | 0.5208 | 0.5283 |
| LTS-SEI | **0.9995** | **0.9971** | **0.9867** | **0.9942** | **0.9650** | **0.9642** |
| Upper Limit | 0.9996 | 0.9992 | 0.9987 | 0.9983 | 0.9992 | 0.9929 |

Note: "A1->A2" indicates that the model trained using Source Domain Data A1 predicts the labels of Target Domain Data A2. There is a time interval between A1 and A2.

*Deep layer*: Deep RFFs refer to output features of the last fully connected layer of the classifier. Table 5 presents the average recognition accuracy of various methods. Compared with the original I/Q signal, center loss (center) can enhance the prediction performance of new data. This indicates that enhancing intraclass consistency can result in correct identification of edge samples in the target domain. HOMM adopts the third-order moment matching loss to train the model described in this study. Compared with the first-order moment matching MMD and CMD, and the second-order moment matching of JDA, HOMM can effectively identify specific emitters in A2, A3, and B6. Therefore, higher-order moments can deeply describe the feature distribution of signals and simplify domain alignment. Additionally, based on the recognition results of A1->A2 and A1->A3, it can be seen that compared with the shallow feature alignment, the existing domain adaptive methods deliver better recognition performances for A2 and A3 when aligning deep

features. This may be because deep features play a decisive role in predicting the labels of specific emitters. However, it is difficult for the existing methods to learn domain-invariant deep features of the long time span-specific emitter signals in Dataset B. Although there may be considerable differences in data distribution between B7, B8, B9 and B1, the proposed LTS-SEI method still enjoys a considerable improvement in identifying two specific emitters.

**Table 5.** Performance comparison between various methods (deep layer).

| Method | Average Recognition Accuracy | | | | | |
|---|---|---|---|---|---|---|
| | **A1->A2** | **A1->A3** | **B1->B6** | **B1->B7** | **B1->B8** | **B1->B9** |
| I/Q | 0.8958 | 0.8592 | 0.9450 | 0.4842 | 0.5758 | 0.6817 |
| Center | 0.9550 | 0.9150 | 0.9533 | 0.4800 | 0.5750 | 0.6808 |
| MMD | 0.9542 | 0.9342 | 0.9467 | 0.5008 | 0.5742 | 0.6883 |
| DAN | 0.9758 | 0.9258 | 0.9517 | 0.4875 | 0.5850 | 0.6617 |
| JDA | 0.9517 | 0.9425 | 0.9483 | 0.4917 | 0.5850 | 0.6483 |
| CMD | 0.9283 | 0.8925 | 0.9575 | 0.4625 | 0.5633 | 0.6500 |
| HOMM | 0.9875 | 0.9575 | 0.9617 | 0.4850 | 0.6242 | 0.7150 |
| LTS-SEI | **0.9995** | **0.9971** | **0.9892** | **0.9567** | **0.9458** | **0.9850** |
| Upper Limit | 0.9996 | 0.9992 | 0.9962 | 0.9983 | 0.9983 | 0.9975 |

*Shallow layer and deep layer*: Additionally, we trained the model to simultaneously align shallow RFFs and deep RFFs simultaneously. Unfortunately, this idea did not achieve better results. For A2 and A3, the existing methods suffered overfitting when the model simultaneously learned domain-invariant shallow features and deep features (see Table 6). Compared with the two feature learning methods previously mentioned, the average recognition accuracy of HOMM on A2 and A3 decreased by 3.09% and 3.42%, 2.5% and 3.33%, respectively. The collection time interval between B14 and B1 is close to one and a half years. The recognition accuracy of the original I/Q signal and existing methods on B14 is near 50%, which means that the model lost its predictive ability. Therefore, even if a high-gain antenna can receive high-quality signals, it is likely to cause considerable deviation in data distribution between the training set and testing set when the time span is long. This may be the result of the simultaneous effect of external environment and internal RFF changes. The same concept applies to other datasets. However, the proposed LTS-SEI method can still extract stable or slowly changing shallow features and deep features, which demonstrates the effectiveness of LTS-SEI in addressing the long time span SEI problem. We suppose that, the satisfactory recognition or prediction performance of LTS-SEI is inseparable from the decent signal quality of Datasets A and B.

### 5.6. Feature Visualization

To intuitively understand the recognition features learned by LTS-SEI, Figure 8 shows the 2D scatter maps of features extracted by different methods for A1->A2, A1->A3, B1->B2, and B1->B3. The first column shows the 2D features of the target domain samples output by the model directly trained using the source domain samples. For A1->A2 and A1->A3, it is evident that some features of Specific Emitters G, H, and J are confused. This indicates that even if the time span between signals is small, the trained model is likely to reduce its predictive ability for new data. For B1->B2 and B1->B3, which have longer time spans, the trained model completely confuses the two specific emitters. The second column displays the 2D features of target domain samples output using the existing domain adaptation methods. The confusion between specific emitters G, H, and J in A1->A2 and A1->A3 was alleviated through domain adaptation. However, the two specific emitters in B1->B2 are still unrecognized, and some samples in B1->B3 are confused. The third column presents the 2D features of the target domain samples output by the proposed LTS-SEI method. Clearly, LTS-SEI effectively separates the features of different specific emitter samples. Thanks to the mutual cooperation between its different components, LTS-SEI can extract RFFs with

interclass separation, intraclass compactness, and domain invariance. Undoubtedly, these RFFs are crucial in identifying the long time span-specific emitter signals.

**Table 6.** Performance comparison between various methods (shallow layer and deep layer).

| Method | Average Recognition Accuracy | | | | | | |
|---|---|---|---|---|---|---|---|
| | A1->A2 | A1->A3 | B1->B10 | B1->B11 | B1->B12 | B1->B13 | B1->B14 |
| I/Q | 0.8958 | 0.8592 | 0.5083 | 0.4917 | 0.7308 | 0.5367 | 0.5150 |
| MMD | 0.9358 | 0.8467 | 0.5108 | 0.5200 | 0.7567 | 0.5558 | 0.5167 |
| DAN | 0.9117 | 0.8742 | 0.5550 | 0.4958 | 0.7892 | 0.5658 | 0.5233 |
| JDA | 0.9133 | 0.8425 | 0.5393 | 0.5092 | 0.7850 | 0.5283 | 0.5067 |
| CMD | 0.9058 | 0.8842 | 0.5358 | 0.5025 | 0.7208 | 0.5433 | 0.4858 |
| HOMM | 0.9533 | 0.9242 | 0.5383 | 0.5208 | 0.7417 | 0.5500 | 0.5025 |
| LTS-SEI | **0.9995** | **0.9971** | **0.9733** | **0.9392** | **0.9942** | **0.9533** | **0.9633** |
| Upper Limit | 0.9996 | 0.9992 | 0.9992 | 0.9979 | 0.9967 | 0.9933 | 0.9971 |

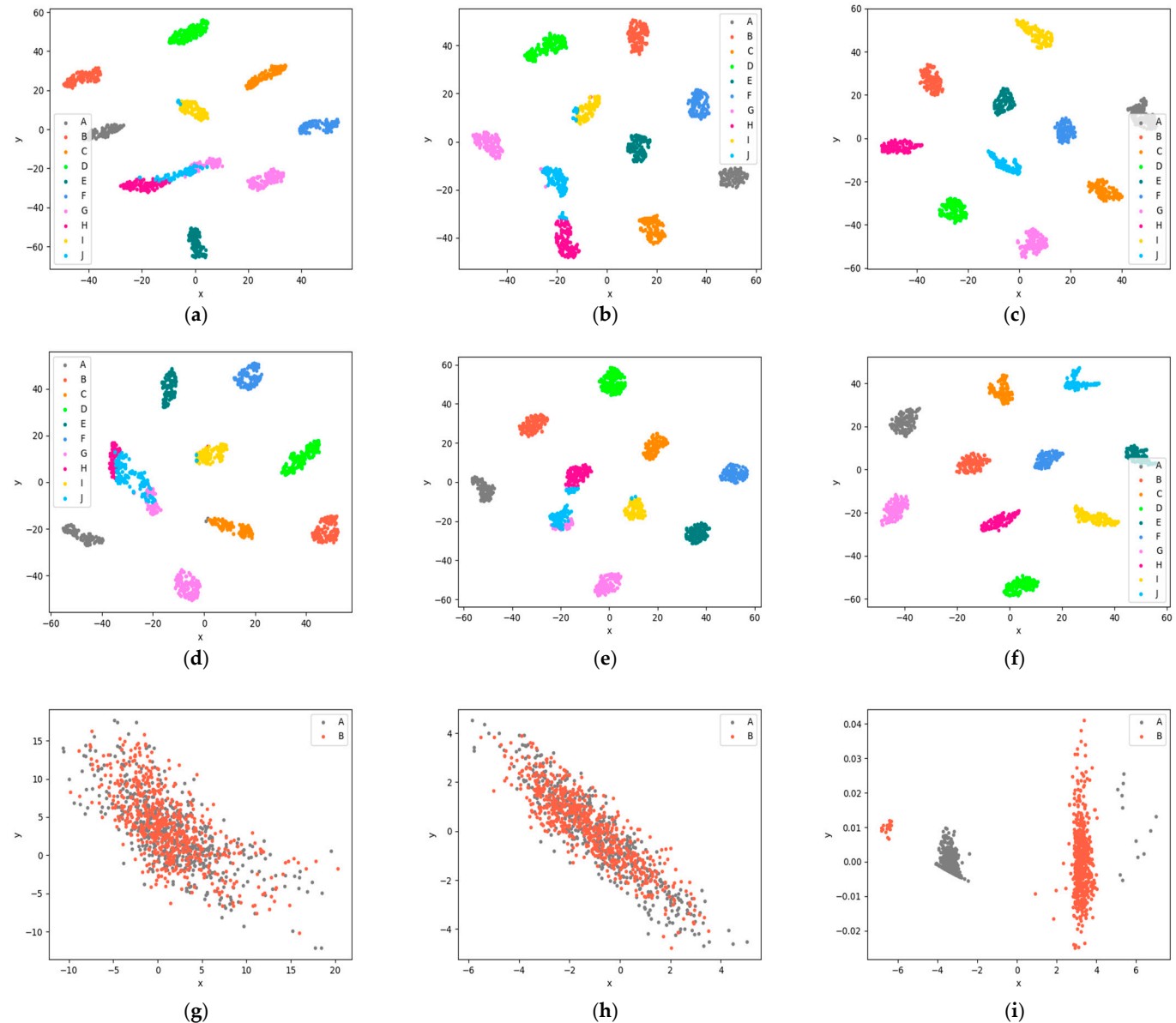

**Figure 8.** *Cont.*

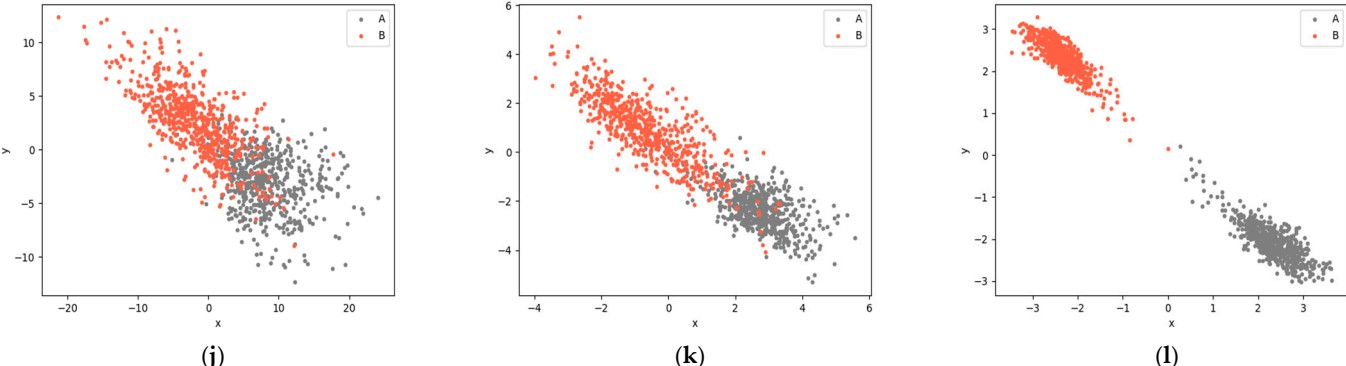

(j)                (k)                (l)

**Figure 8.** 2D feature scatter maps of the output features of the last fully connected layer of the classifier after t-distributed stochastic neighbor embedding dimensionality reduction: (**a**) A1->A2 (I/Q), (**b**) A1->A2 (shallow layer: CMD), (**c**) A1->A2 (LTS-SEI), (**d**) A1->A3 (I/Q), (**e**) A1->A3 (deep layer: HOMM), (**f**) A1->A3 (LTS-SEI), (**g**) B1->B2 (I/Q), (**h**) B1->B2 (shallow layer and deep layer: MMD), (**i**) B1->B2 (LTS-SEI), (**j**) B1->B3 (I/Q), (**k**) B1->B3 (shallow layer and deep layer: JDA), and (**l**) B1->B3 (LTS-SEI).

## 5.7. Specific Emitter Identification Results

Based on the 2D feature scatter maps described in Section 5.6, we present the corresponding confusion matrix in Figure 9. Similar to the abovementioned analysis, specific emitter J in A1->A2 and A1->A3 is easily identified as G. A small sample of specific emitter H is recognized as J. Specific emitters G, H, and J all come from the same navigation satellite system and the signal formats they transmit are exactly the same, which increases the difficulty of identifying similar individuals. For B1->B2, 57% of the samples from Specific Emitter A are identified as from B, which indicates that A and B are completely inseparable. Although the situation of B1->B3 is relatively optimistic, nearly one-third of the samples from Specific Emitter B are classified as from A. The existing domain adaptive methods suffer limited ability to solve the sample confusion problem. They still cannot provide reliable identification results for some long time span datasets, such as B1->B2 (with a considerable difference between the source domain and target domain). However, even if long time span presents considerable challenges, LTS-SEI can still identify different specific emitters with considerable accuracy. We can see that the classification accuracy of all specific emitter samples is nearly 100%, as shown in the third column of Figure 9. This predictive ability is crucial for practical applications when the labels of new data are not known.

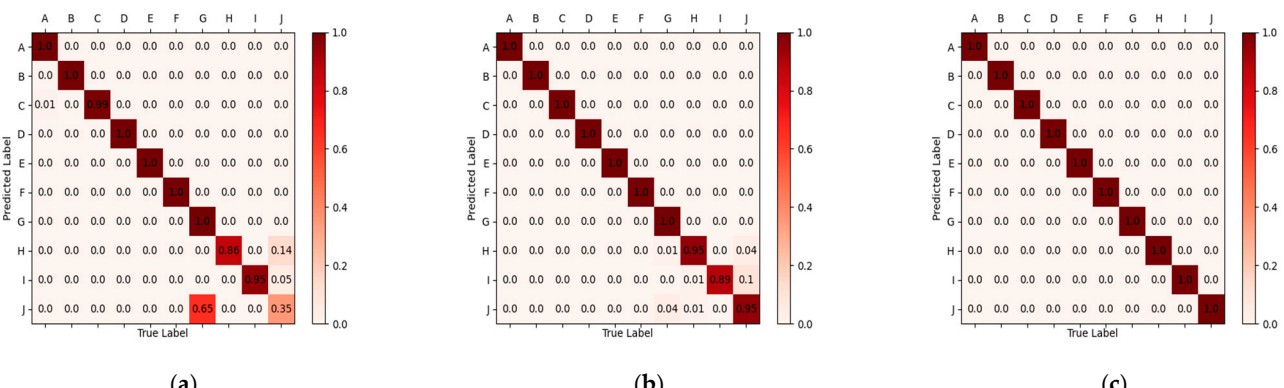

(a)                (b)                (c)

**Figure 9.** *Cont.*

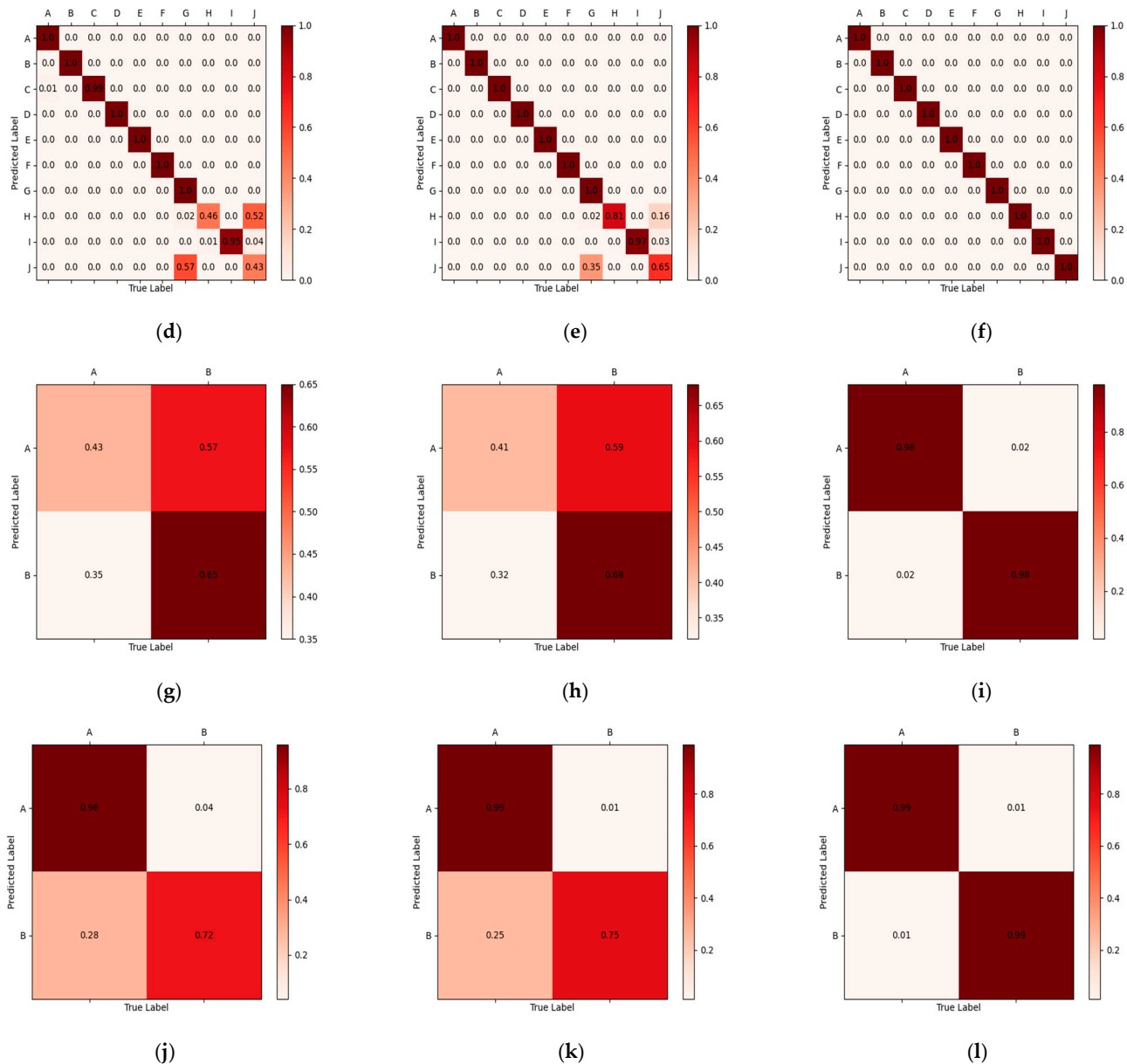

**Figure 9.** Confusion matrix for identification of different specific emitters: (**a**) A1->A2 (I/Q), (**b**) A1->A2 (Shallow layer: CMD), (**c**) A1->A2 (LTS-SEI), (**d**) A1->A3 (I/Q), (**e**) A1->A3 (Deep layer: HOMM), (**f**) A1->A3 (LTS-SEI), (**g**) B1->B2 (I/Q), (**h**) B1->B2 (Shallow layer and deep layer: MMD), (**i**) B1->B2 (LTS-SEI), (**j**) B1->B3 (I/Q), (**k**) B1->B3 (Shallow layer and deep layer: JDA), and (**l**) B1->B3 (LTS-SEI).

### 5.8. Training Accuracy and Loss of LTS-SEI

Figures 10 and 11 show the training accuracy curves and loss curves on A1->A2, A1->A3, B1->B2, B1->B6, B1->B9, and B1->B14, respectively, to more effectively understand the samples training process of LTS-SEI. From Figure 10, it can be seen that the classification accuracy of source domain samples and target domain samples increases with increase of iterations. For A2 and A3, which have a shorter time span, improvement in their classification accuracy is almost synchronous with A1. For Dataset B, which has a long time span, when the data distribution of the source domain samples and the target domain samples is similar (B1->B6), their training accuracy quickly converges. The source domain samples with available labels train faster than the target domain samples without labels when there is a considerable difference between the source domain and target domain (B1->B2, B1->B9, and B1->B14). The improvement in target domain samples classification accuracy cannot

be achieved without the joint efforts of various components of the LTS-SEI framework, including the confrontation between the feature extractor and domain discriminator, the alignment of deep features, and the correct recognition of edge samples. From Figure 11, it can be seen that the training loss of the source domain samples continuously decreases with the increasing of iterations and eventually converges to approximately 0. The adversarial loss between the feature extractor and domain discriminator can ultimately achieve the Nash equilibrium, which aligns the shallow and deep features of the source domain samples and target domain samples in the feature space. The training accuracy curves and training loss curves indicate that LTS-SEI has the ability to learn RFFs of target domain samples that remain unchanged over an extended time span.

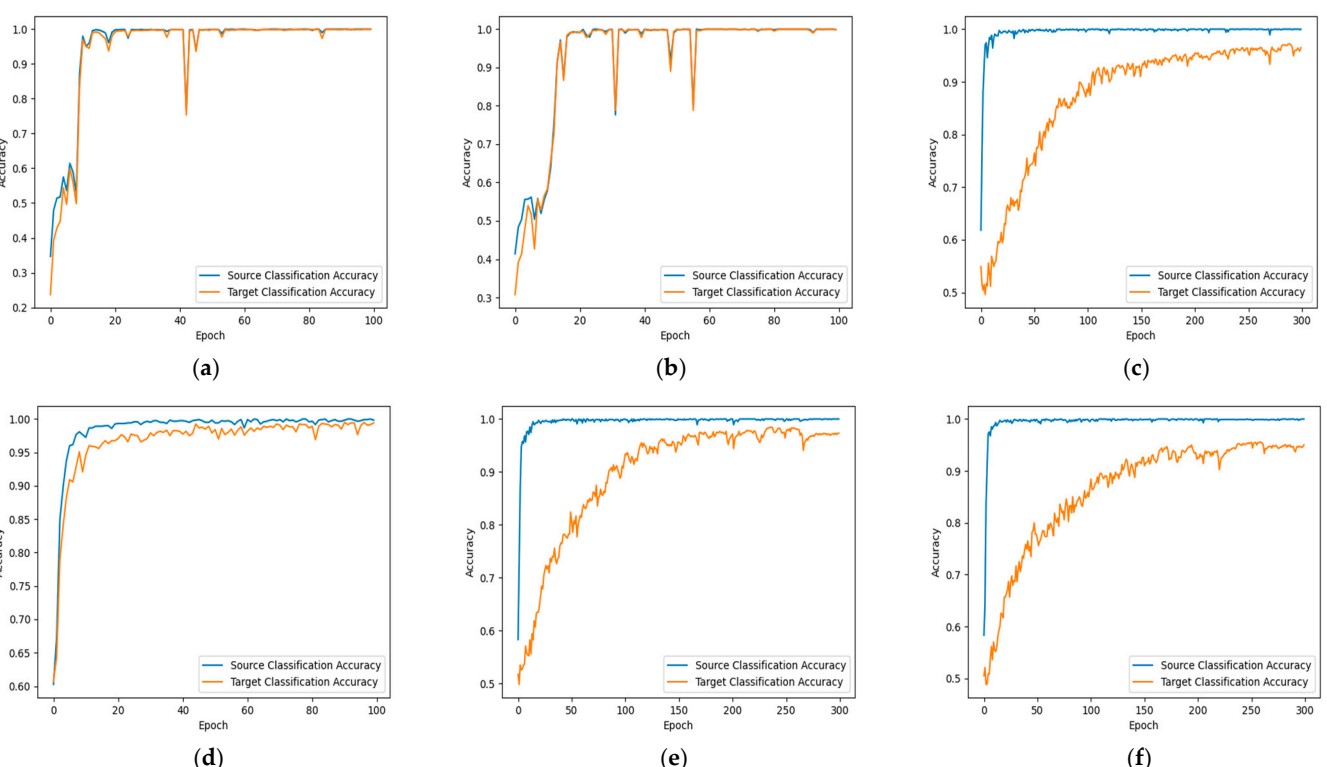

**Figure 10.** Training accuracy curves of LTS-SEI on (**a**) A1->A2, (**b**) A1->A3, (**c**) B1->B2, (**d**) B1->B6, (**e**) B1->B9, and (**f**) B1->B14.

*5.9. Ablation Experiment*

An ablation experiment is used to explore the impact of various components of the LTS-SEI framework on the recognition performance of long time span-specific emitter signals. Although some experimental data are duplicated from Section 5.5, we must present them again to evaluate the contribution of every component of the LTS-SEI framework to the recognition performance. Tables 7–10 present six ablation studies: (1) Original I/Q, (2) Shallow confrontation, (3) Deep alignment, (4) LTS-SEI (no Center Loss), (5) LTS-SEI (no HOMM3 Loss), and (6) LTS-SEI. Evidently, the proposed LTS-SEI method achieves the optimum results in terms of accuracy, precision, recall, and F1-score on multiple datasets. Shallow confrontation can enhance the recognition effect of specific emitters to a certain extent. On A1->A2 and A1->A3, deep alignment seems to have overfitting after incorporating Center Loss, which results in its recognition accuracy being lower than the previously mentioned deep layer HOMM domain adaptive method. When the Center Loss or HOMM3 Loss of LTS-SEI does not participate in model training, the accuracy, precision, recall, and F1-score on most datasets will decrease. For example, incorporating Center Loss can increase the accuracy of the model by up to 2.09%, and incorporating HOMM3 Loss by 41.41%. Additionally, by comparing shallow confrontation and deep alignment, we can

observe that with adversarial learning, it is easier to obtain domain-invariant RFFs for real data with dynamic changes. However, better results cannot be achieved solely through shallow confrontation and deep alignment. Only when different components of LTS-SEI work together to learn domain-invariant shallow and deep fingerprints can the accuracy of the long time span SEI be considerably increased.

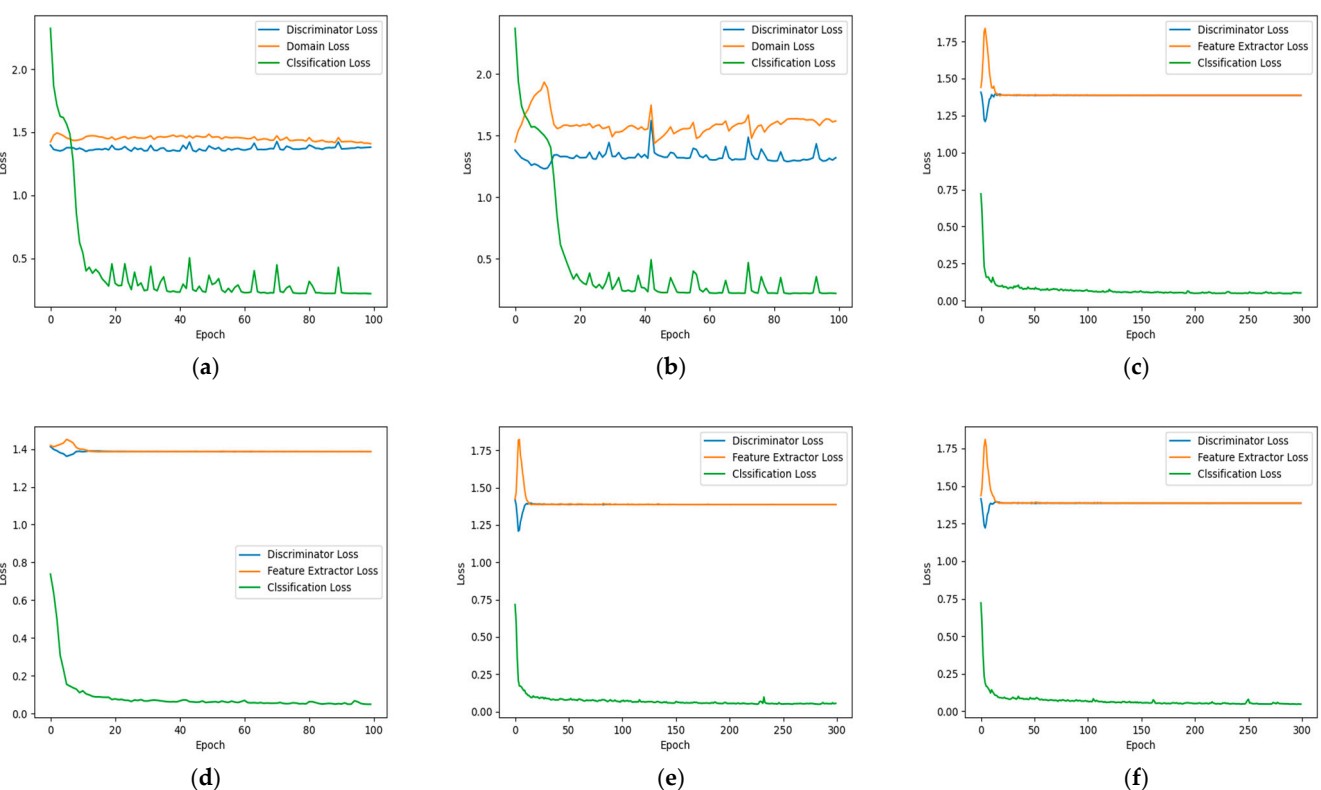

**Figure 11.** Training loss curves of LTS-SEI on (**a**) A1->A2, (**b**) A1->A3, (**c**) B1->B2, (**d**) B1->B6, (**e**) B1->B9, and (**f**) B1->B14.

*5.10. Expansion to Small Training Samples*

We introduced sample proportion to adjust the number of training samples in a bid to further explore the identification performance of the proposed LTS-SEI method when extended to a small number of training samples. Noteworthily, the sample proportion is defined as the ratio of the number of samples in the training set to those in the validation set. The samples of B1 are assumed to come from the source domain. The samples of B2, B5, B7, B9, B12, and B14 come from the target domain. The number of training samples in the source domain and target domain increases with increasing sample proportion. It is obvious that the greater the sample proportion, the better the recognition performance of LTS-SEI on all data subsets, as shown in Figure 12. Due to the same navigation signal formats transmitted by the two GPS satellites, LTS-SEI confuses different samples when the number of training samples is small. However, the predictive ability of LTS-SEI for new data from B2, B5, B7, and B14 considerably deteriorates when the sample proportion is less than 1.25. The recognition accuracy of LTS-SEI on all data subsets approaches saturation when the sample proportion is 3.5.



**Table 7.** Ablation experiment (accuracy).

| Dataset | Component | | | | | |
|---|---|---|---|---|---|---|
| | Original I/Q | Shallow Confrontation | Deep Alignment | LTS-SEI (No Center Loss) | LTS-SEI (No HOMM3 Loss) | LTS-SEI |
| A1->A2 | 0.8958 | 0.9975 | 0.9375 | 0.9975 | 0.9992 | **0.9995** |
| A1->A3 | 0.8592 | 0.9925 | 0.8667 | 0.9946 | 0.9942 | **0.9971** |
| B1->B2 | 0.5200 | 0.5550 | 0.5708 | 0.9658 | 0.5742 | **0.9867** |
| B1->B3 | 0.8650 | 0.9425 | 0.8392 | **0.9992** | 0.9217 | 0.9942 |
| B1->B9 | 0.6817 | 0.6942 | 0.6800 | 0.9800 | 0.6958 | **0.9850** |
| B1->B14 | 0.5150 | 0.4950 | 0.4908 | 0.9425 | 0.5492 | **0.9633** |

**Table 8.** Ablation experiment (Precision).

| Datas Set | Component | | | | | |
|---|---|---|---|---|---|---|
| | Original I/Q | Shallow Confrontation | Deep Alignment | LTS-SEI (No Center Loss) | LTS-SEI (No HOMM3 Loss) | LTS-SEI |
| A1->A2 | 0.9410 | 0.9976 | 0.9450 | 0.9975 | 0.9992 | **0.9995** |
| A1->A3 | 0.8905 | 0.9931 | 0.8961 | 0.9946 | 0.9942 | **0.9971** |
| B1->B2 | 0.5190 | 0.5546 | 0.5710 | 0.9658 | 0.5757 | **0.9867** |
| B1->B3 | 0.8909 | 0.9446 | 0.8732 | **0.9992** | 0.9277 | 0.9942 |
| B1->B9 | 0.7097 | 0.7069 | 0.6969 | 0.9806 | 0.6976 | **0.9851** |
| B1->B14 | 0.5167 | 0.4956 | 0.4898 | 0.9441 | 0.5491 | **0.9633** |

**Table 9.** Ablation experiment (recall).

| Dataset | Component | | | | | |
|---|---|---|---|---|---|---|
| | Original I/Q | Shallow Confrontation | Deep Alignment | LTS-SEI (No Center Loss) | LTS-SEI (No HOMM3 Loss) | LTS-SEI |
| A1->A2 | 0.8958 | 0.9975 | 0.9375 | 0.9975 | 0.9992 | **0.9995** |
| A1->A3 | 0.8592 | 0.9925 | 0.8667 | 0.9946 | 0.9942 | **0.9971** |
| B1->B2 | 0.5200 | 0.5550 | 0.5708 | 0.9658 | 0.5742 | **0.9867** |
| B1->B3 | 0.8650 | 0.9425 | 0.8392 | **0.9992** | 0.9217 | 0.9942 |
| B1->B9 | 0.6817 | 0.6942 | 0.6800 | 0.9800 | 0.6958 | **0.9850** |
| B1->B14 | 0.5150 | 0.4950 | 0.4808 | 0.9425 | 0.5492 | **0.9633** |

**Table 10.** Ablation experiment (F1-score).

| Dataset | Component | | | | | |
|---|---|---|---|---|---|---|
| | Original I/Q | Shallow Confrontation | Deep Alignment | LTS-SEI (No Center Loss) | LTS-SEI (No HOMM3 Loss) | LTS-SEI |
| A1->A2 | 0.8703 | 0.9975 | 0.9370 | 0.9975 | 0.9992 | **0.9995** |
| A1->A3 | 0.8490 | 0.9925 | 0.8495 | 0.9946 | 0.9942 | **0.9971** |
| B1->B2 | 0.5190 | 0.5538 | 0.5660 | 0.9658 | 0.5686 | **0.9867** |
| B1->B3 | 0.8634 | 0.9425 | 0.8362 | **0.9992** | 0.9215 | 0.9942 |
| B1->B9 | 0.6742 | 0.6909 | 0.6752 | 0.9800 | 0.6957 | **0.9850** |
| B1->B14 | 0.5148 | 0.4950 | 0.4898 | 0.9424 | 0.5491 | **0.9633** |

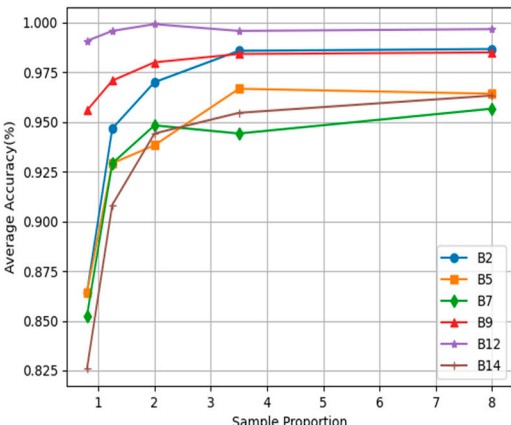

**Figure 12.** Identification result on small training samples.

## 6. Conclusions

This study proposed an unsupervised method called LTS-SEI, which is the first work to address the long time span SEI problem. Domain adaptation was introduced into SEI to learn stable RFFs that are less affected by time changes. On the one hand, LTS-SEI implements adversarial learning at the shallow layer and aligns features at the deep layer of the model, which enables it to learn domain-invariant RFFs. On the other hand, LTS-SEI enhances intraclass consistency of deep features, which effectively increases the accuracy of SEI. The idea proved feasible in solving the long time span SEI problem. We demonstrated the effectiveness of LTS-SEI from different perspectives using real received satellite navigation signals. When the existing methods suffer difficulty in identifying long time span signals, LTS-SEI can still recognize specific emitter signals with time intervals approaching 2 years. This indicates that LTS-SEI has practical ability to predict the identity of specific emitters. However, the following three topics on long time span SEI need further research:

(1) In this study, we used 13-m and 40-m large-aperture antennas to receive space signals, which enhanced the quality of the signals. However, new methods need to be studied when addressing low SNR-specific emitter signals with long time spans.

(2) The experimental data in this study are of high-orbit satellite signals, with the sampling rate of the acquisition equipment being 250 MHz. Sufficient data samples ensure effective training of deep models. However, we must develop small samples for long time span SEI methods when the number of training samples is small.

(3) In practice, the model's predictive ability must be based on real training data without labels. Therefore, we must design new unsupervised learning algorithms to further enhance the practicality of long time span SEI.

**Author Contributions:** Conceptualization, P.L.; methodology, P.L. and L.G.; software, P.L. and H.Z.; validation, P.L.; formal analysis, P.L., P.S. and Z.C.; investigation, P.L. and L.G.; resources, L.G., H.Z. and X.L.; data curation, H.Z.; writing—original draft preparation, P.L.; writing—review and editing, P.L., L.G., P.S. and Z.C.; visualization, P.L.; supervision, L.G. and X.L.; project administration, L.G. and H.Z.; funding acquisition, L.G. and H.Z. All authors have read and agreed to the published version of the manuscript.

**Funding:** This work was supported by the Technical Support Talent Plan of Chinese Academy of Science (Grant No. E317YR17), the Project for Guangxi Science and Technology Base and Talents (Grant No. GK AD22035957), the National Natural Science Foundation of China (Grant No. 12273045), the Western Talent Introduction Project of Chinese Academy of Sciences (Grant No. E016YR1R) and High Level Talent Project of Shaan xi Province (Grant No. E039SB1K).

**Data Availability Statement:** Not applicable.

**Conflicts of Interest:** The authors declare no conflict of interest.

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
