# Peer review of "A Long Time Span-Specific Emitter Identification Method Based on Unsupervised Domain Adaptation"

_remotesensing, doi:10.3390/rs15215214_

Round 1

Reviewer 1 Report

Comments and Suggestions for Authors

Comments on the Quality of English Language

Reviewer 2 Report

Comments and Suggestions for Authors

Review of Remote Sensing Manuscript ID: 2639718

A Long Time Span Specific Emitter Identification Method Based on Unsupervised Domain Adaptation

The authors of this paper address the challenge specific emitter identification (SEI), i.e., recognizing the specific source of emitted radio signals based on the characteristic features in the signal. This is of prime importance in multiple application areas such as recognizing illegal use of licensed spectrum bandwidths, identifying malicious users of bandwidths, recognizing friendly/unfriendly sources in a military combat environment, etc. A key issue with evaluation of methods is the lag and associated difference in data distribution between training and test sets. Furthermore, obtaining labels for supervised evaluation is difficult. The authors propose to address the long timespan (LTS) SEI problem with an unsupervised domain adaptive method they call LTS-SEI. The first stage of LTS-SEI is a feature extractor. A discriminator that classifies whether the feature belongs to the source or target domain incentivizes the extractor to generate domain adaptive features and thereby make the learned representations domain invariant. At the same time, a classifier achieves domain alignment by applying feature matching between source domain and target domain samples. A training objective with three components is proposed: a softmax cross-entropy classification loss, a feature center loss for clustering, and a higher order moments based HOMM3 loss. This reviewer finds the paper to be within the scope of Remote Sensing and an important contribution. However, this reviewer has a few questions regarding the methodology of the study. This reviewer recommends that this paper be revised addressing this queries before being accepted for publication.

Strengths:

1.     Paper is well-written, well-motivated and has appropriate review of the literature for context.

2.     The proposed model architectures, training procedures, and experiments are appropriately described.

3.     The proposed method has been compared to existing methods.

Questions and Comments:

1.     Please describe what the labels are on line 184? Possibly, take an example and demonstrate the problem setting. It is not clear to this reviewer what the labels are for the classification problem.

2.     What happens if we replace the center loss with say a contrastive loss such as SimCLR’s NT-XEnt loss? Or maybe a SWaV-based Sinkhorn-Knopp clustering method? Why did the authors choose the specific form of the center loss?

3.     There is no ablation study provided to understand the relative importance of the 3 loss components. Please provide a detailed ablation study on the loss components weights.

Reviewer 3 Report

Comments and Suggestions for Authors

This study proposes an unsupervised method to address the long time span specific emitter identification (SEI) problem. The experimental section is adequate and the structure of the manuscript is reasonable, but the following issues should be addressed.

1.The authors summarizes the existing SEI methods in detail, but still recommends explaining the specific advantages of deep learning applied to SEI.

2. The authors should clarify in the introduction what is the significance of studying the SEI with a longer signal span? At the same time, it is necessary to explain how the time span affects the similarity of data distribution between the training set and the test set.

3. Will different results be obtained when the number of samples of training set, verification set and test set are divided according to other ratio?

4. The authors should explain why different models are used for different data sets, as it is difficult to guarantee the generalization ability of the proposed method.

5. The authors should add some more reasonable accuracy evaluation methods.

6. Regarding the limitations of the proposed method as well as directions for improvement and optimization should be discussed.

7. The English language needs polishing.

Line 684: “proposes a unsupervised” should be “an”.

Comments on the Quality of English Language

Moderate editing of English language required

Round 2

Reviewer 1 Report

Comments and Suggestions for Authors

I have no more questions.

Reviewer 2 Report

Comments and Suggestions for Authors

This reviewer is satisfied with the revisions made by the authors.

Reviewer 3 Report

Comments and Suggestions for Authors

The manuscript has been appropriately revised and can therefore be accepted.